# SAMPLE EFFICIENT REWARD AUGMENTATION IN OFFLINE-TO-ONLINE REINFORCEMENT LEARNING

## ABSTRACT

Offline-to-online RL can make full use of pre-collected offline datasets to initialize policies, resulting in higher sample efficiency and better performance compared to only using online algorithms alone for policy training. However, direct fine-tuning of the pre-trained policy tends to result in sub-optimal performance. A primary reason is that conservative offline RL methods diminish the agent's capability of exploration, thereby impacting online fine-tuning performance. In order to encourage agent's exploration during online fine-tuning and enhance the overall online fine-tuning performance, we propose a generalized reward augmentation method called **S**ample **E**fficient **R**eward **A**ugmentation (**SERA**). Specifically, SERA encourages agent to explore by computing Q conditioned entropy as intrinsic reward. The advantage of SERA is that it can extensively utilize offline pre-trained Q to encourage agent uniformly coverage of state space while considering the imbalance between the distributions of high-value and low-value states. Additionally, SERA can be effortlessly plugged into various RL algorithms to improve online fine-tuning and ensure sustained asymptotic improvement. Moreover, we conducted extensive experiments using SERA and found that SERA significantly improves CQL (**21**%) and Cal-QL (**11.2**%). Simultaneously, we further extended the experimental tests to other model-free algorithms, and the results demonstrate the generality of SERA.

## 1 INTRODUCTION

Offline reinforcement learning (RL) holds a natural advantage over online RL in that it can be completely trained using pre-existing static datasets, obviating the necessity to interact with the environment for the collection of new trajectories (Levine et al., 2020). Nevertheless, offline RL faces limitations due to it tends to learn the sub-optimal performance if the action support can't be well estimated, and also risk of overestimating out-of-distribution (OOD) state actions. Consequently, it becomes imperative to address these limitations by enhancing

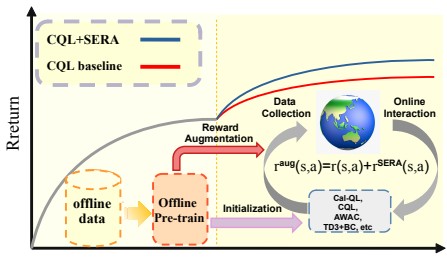

Figure 1: Demonstration of SERA.

the performance of the offline policy through the process of online fine-tuning (Fujimoto & Gu, 2021; Kostrikov et al., 2021; Wu et al., 2022; Mark et al., 2023).

Drawing inspiration from fine-tuned based modern machine learning, which leverages pre-training followed by fine-tuning on downstream tasks (Brown et al., 2020; Touvron et al., 2023), it seems plausible to elevate the performance of offline policies through the process of online fine-tuning. However, previous studies demonstrate that offline pre-trained policy tends to exhibit worse fine-tuning performance. In particular, the offline initialized policy especially suffer from performance drop during the early online stage, which is caused by distribution shift and overestimation of OOD state actions (Nakamoto et al., 2023) or the problem of misaligned value estimation in online and offline training stage (Nair et al., 2021).

To address above limitations, an effective method is to firstly pre-train on offline dataset with offline algorithm and following by utilizing exploratory policy (*Approach* 1) to conduct online fine-tuning,

or aligning the value estimation in online and offline thereby enabling online fine-tuning without performance decreasing (*Approach* 2). Specifically, *Approach* 1 utilize pessimistic offline RL methods for pre-training while incorporating exploration into online fine-tuning (Lee et al., 2021a; Mark et al., 2023; Wu et al., 2022). However, when directly fine-tuning the offline pre-trained policy, there exhibit performance drop at the early fine-tuning stage. *Approach* 2 aims to address the limitation of *Approach* 1 by calibrated method (Nakamoto et al., 2023) that is learning a better initialization thus enabling standard online fine-tuning by aligning the value estimation in offline and online stage. Nonetheless, *Approach* 2 still is cooperated with exploratory policy [1]. Thus, both *Approach* 1 and *Approach* 2 use policy or methods that are exploratory in nature, and therefore, keeping the agent exploratory seems to be the key to ensure offline-to-online performance. Therefore, can we improve offline-to-online by only enhancing exploration?

We hypothesis that it's feasible to improve offline-to-online by only enhancing exploration, because as long as an agent can quickly and uniformly explore the observation space, it can collect more diverse dataset, the collected dataset helps to mitigate the shortcomings of the conservative policy (Luo et al., 2023). Meanwhile, collected dataset also helps to alleviate the overestimation of OOD state actions and recover the real value estimation thereby achieving better fine-tuning performance. Based on such insight, we propose a generalized offline-to-online framework called **S**ample **E**fficient **R**eward **A**ugmentation (**SERA**), which encouraging offline pre-trained policy to explore by computing Q conditioned state entropy as intrinsic reward. Specifically, as shown in Figure 1, our SERA mainly has two phages, which firstly pre-trained policy with model-free algorithm and followed by online fine-tuing with reward augmentation. In particular, we utilize offline pre-trained Q network to compute value conditioned entropy as intrinsic reward, which benefits from both high sample efficiency and fine-tuned performance by encouraging the agent to explore the observation space uniformly across different values. Compared with previous offline-to-online methods, SERA has various advantages: 1) **Adaptability.** Different from regularized-based or supported constraint methods adding term to constraint policy, SERA can be seamlessly plugged into various model-free offline algorithms to conduct offline-to-online RL, thereby getting ridding of the limitations [2] of supported or regularized based method. 2) **Pluggable and Flexibility.** Different from most of offline-to-online methods, our SERA can be paired with most of existing offline RL methods and improving their fine-tuning performance.

To summarize, our contribution can be summarized as follows:

- Firstly, we propose a generalized reward augmentation framework that can be plugged into various offline algorithms to conduct offline-to-online setting and improve their online fine-tuning performance.

- Secondly, compared with previous state entropy maximization methods, we utilize $Q$ conditional state entropy as intrinsic reward, thus can decrease the biased exploration by considering the imbalance distribution of value space of decision makings.

- Lastly, we also provided mathematics analysis to prove that SERA can provide guarantees to monotonic policy improvement of soft $Q$ optimization (Haarnoja et al., 2018), and conservative policy improvement (theorem 4.2).

## 2 RELATED WORK

**Offline RL.** The notorious challenge within offline RL pertains to the mitigation of out-of-distribution (OOD) predictions, which are a consequence of the distributional shift between the behavior policy and the training policy (Fujimoto et al., 2019b). To effectively address this issue, **1)** conservative policy-based *model-free* methods adopt the following approaches: Adding policy regularization (Fujimoto et al., 2019a; Kumar et al., 2019; Wu et al., 2019; Liu et al., 2023b), or implicit policy constraints (Peng et al., 2019; Siegel et al., 2020; Zhou et al., 2020; Chen et al., 2022; Wu et al., 2022; Liu et al., 2023b;a; Zhuang et al., 2023). **2)** And, conservative critic-based model-free methods penalize the value estimation of OOD state-actions via conducting pessimistic

---

[1]CQL has two variants, which including CQL-DQN and CQL-SAC. In particular, CQL-SAC is based on SAC (Haarnoja et al., 2018), which is maximum entropy policy with highly exploratory.

[2]Regularized or support constraint method have to estimate action support, thus policy learning will be affected if the action support can't be well estimated.

Q function (Kumar et al., 2020a) or uncertainty estimation (An et al., 2021; Bai et al., 2022; Reza-eifar et al., 2022; Wu et al., 2021) or implicitly regularizing the bellman equation (Kumar et al., 2020b; Liu et al., 2022). In terms of the *model-base* offline RL, it similarly train agent with distribution regularization (Hishinuma & Senda, 2021; Yang et al., 2022; Zhang et al., 2022), uncertainty estimation (Yu et al., 2020; Kidambi et al., 2020; Lu et al., 2022), and value conservation (Yu et al., 2021). In our research, due to the remarkable sampling efficiency and outstanding performance of model-free algorithms in both offline and online RL settings, and we prove that SERA satisfy the guarantee of Soft-Q optimization (theorem 4.1), thus we select *Conservative Q-Learning* (CQL) and *Calibrated Q-Learning* (Cal-QL) as our primary baseline methods. Additionally, to conduct a thorough assessment of the effectiveness of our proposed approaches, we have also expanded our evaluation to encompass a diverse set of other model-free algorithms, including *Soft-Actor-Critic* (SAC) (Haarnoja et al., 2018), *Implicit Q-learning* (IQL) (Kostrikov et al., 2021), *TD3+BC* (Fujimoto & Gu, 2021), and *AWAC* (Nair et al., 2021).

**Offline-to-Online RL.** Previous researches have demonstrated that offline RL methods offer the potential to expedite online training, a process that involves incorporating offline datasets into online replay buffers (Nair et al., 2021; Vecerik et al., 2018; Todd Hester & et al., 2017) or initializing the pre-trained policy to conduct online fine-tuning (Kostrikov et al., 2021; Beeson & Montana, 2022). However, there exhibits worse performance when directly fine-tuning the offline pre-trained policy (Nakamoto et al., 2023; Lee et al., 2021b), and such an issue can be solved by adapting a balanced replay scheme aggregated with pessimistic pre-training (Lee et al., 2021b), or pre-training with pessimistic Q function and fine-tuning with exploratory methods (Wu et al., 2022; Mark et al., 2023; Nakamoto et al., 2023). In particular, our approach SERA differs from these methods in that it enhances online fine-tuning solely by augmenting online exploration.

**Online Exploration.** Recent advances in the studies of exploration can obviously improve the online RL sample efficiency, among that, remarkable researches include injecting noise into state actions(Lillicrap et al., 2019) or designing intrinsic reward by counting visitation or errors from predictive models (Badia et al., 2020; Sekar et al., 2020; Whitney et al., 2021; Burda et al., 2018). In particular, the approaches most related to our study are to utilize state entropy as an intrinsic reward (Kim et al., 2023; Seo et al., 2021).

## 3 PRELIMINARY

We formulate RL as Markov Decision Process (MDP) tuple *i.e.*, $\mathcal{M} = (\mathcal{S}, \mathcal{A}, r, T, p_0, \gamma)$. Specifically, $p_0$ denotes the initial state distribution, $\mathcal{S}$ denotes the observation space, $\mathcal{A}$ denotes the actions space, $r : \mathcal{S} \times \mathcal{A} \mapsto \mathbb{R}$ denotes the reward function, $T : \mathcal{S} \times \mathcal{A} \times \mathcal{S}$ denotes the transition function (dynamics), and $\gamma \in [0, 1]$ denotes discount factor. The goal of RL is to find or obtain an optimal policy $\pi^* : \mathcal{S} \mapsto \mathcal{A}$ to maximize the accumulated discounted return *i.e.*, $\pi^* = \arg\max_\pi \mathbb{E}_{\tau \sim \pi(\tau)}[R(\tau)]$, where $\mathbb{E}_{\tau \sim \pi(\tau)}[R^\pi(\tau)] = \mathbb{E}[\sum_{t=0}^\infty \gamma^t r(\mathbf{s}_t, \mathbf{a}_t) | \mathbf{s}_0 \sim p_0, \mathbf{a}_t \sim \pi(\cdot|\mathbf{s}_t), \mathbf{s}_{t+1} \sim T(\cdot|\mathbf{s}_t, \mathbf{a}_t)]$, and $\tau = \{\mathbf{s}_0, \mathbf{a}_0, r_0, \cdots, \mathbf{s}_N, \mathbf{a}_N, r_N\}$ is the rollout trajectory. We also define Q function by $Q^\pi(\mathbf{s}, \mathbf{a}) = \mathbb{E}_{\tau \sim \pi(\tau)}[\sum_{t=0}^T \gamma^t r(\mathbf{s}_t, \mathbf{a}_t) | \mathbf{s}_0 = \mathbf{s}, \mathbf{a}_0 = \mathbf{a}]$, and value function by $V^\pi(\mathbf{s}) = \mathbb{E}_{\mathbf{a} \sim \pi(\mathbf{a}|\mathbf{s})}[Q^\pi(\mathbf{s}, \mathbf{a})]$. Furthermore, in offline-to-online RL problem setting, the agent has to access the static datasets $\mathcal{D}_{\text{offline}}$ for pre-training, followed by conducting online fine-tuning. In this research, we mainly focusing on improving model-free algorithms to conduct offline-to-online RL setting.

**Model-free Offline RL.** Typically, model-free RL algorithms alternately optimize policy with Q-network *i.e.*, $\pi := \arg\max_\pi \mathbb{E}_{\mathbf{s} \sim \mathcal{D}, \mathbf{a} \sim \pi(\cdot|\mathbf{s})}[Q^\pi(\mathbf{s}, \mathbf{a})]$, and conduct policy evaluation by the Bellman equation iteration *i.e.*, $Q^\pi \leftarrow \arg\min_Q \mathbb{E}_{(\mathbf{s}, \mathbf{a}, \mathbf{s}') \sim \mathcal{D}}[(Q^\pi(\mathbf{s}, \mathbf{a}) - \mathcal{B}_\mathcal{M} Q^\pi(\mathbf{s}, \mathbf{a}))^2]$, where $\mathcal{B}_\mathcal{M}^\pi Q(\mathbf{s}, \mathbf{a}) = r(\mathbf{s}, \mathbf{a}) + \gamma \mathbb{E}_{\mathbf{s}' \sim \mathcal{D}}[Q(\mathbf{s}', \pi(\cdot|\mathbf{s}'))]$. In particular, model-free offline RL aims to learn from the static RL datasets $\mathcal{D}_{\text{offline}}$ collected by behavior policy $\pi_\beta$ without access to the environment for collecting new trajectories, therefore, it always suffer from the out-of-distribution (OOD) issues. Specifically, model-free algorithms train the Q function by one step bellman equation *i.e.*, $\mathcal{J}(Q) = \mathbb{E}_{(\mathbf{s}, \mathbf{a}, \mathbf{s}') \sim \mathcal{D}}[(Q^\pi(\mathbf{s}, \mathbf{a}) - \mathcal{B}_\mathcal{M} Q^\pi(\mathbf{s}, \mathbf{a}))^2]$ which requires computing $\mathcal{B}_\mathcal{M}^\pi Q(\mathbf{s}, \mathbf{a}) = r(\mathbf{s}, \mathbf{a}) + \gamma Q(\mathbf{s}', \pi(\cdot|\mathbf{s}'))$, but if $(\mathbf{s}', \pi(\cdot|\mathbf{s}')) \notin \mathcal{D}_{\text{offline}}$ then the overestimation of OOD state actions will cause extrapolation error and learned biased $Q$ further affect $\pi$. Previous studies have extensively studied such a problem, Kumar et al. (2020a) proposed penalizing OOD state actions

by conservative term, and IQL (Kostrikov et al., 2021) implicitly learns Q function with expected regression without explicitly access to the value estimation of OOD state-actions.

Before formally proposing SERA, we firstly define various fundamental concepts:

**Definition 1 (Marginal State distribution)** *Given the trajectory of current empirical policy:* $\tau \sim \pi(\tau)$, *We define state marginal distribution of current empirical policy as:* $\rho_\pi(\mathbf{s}) = \mathbb{E}_{\mathbf{s}_0 \sim p_0, \mathbf{a}_t \sim \pi_\theta(\cdot|\mathbf{s}_t), \mathbf{s}_{t+1} \sim T(\cdot|\mathbf{s}_t, \mathbf{a}_t)}[\frac{1}{N} \sum_{t=1}^{N} \mathbb{1}(\mathbf{s}_t = \mathbf{s})]$.

**Definition 2 (Conditional Entropy)** *Given two discrete random variables X and Y with a joint probability mass function denoted as P(X, Y), the marginal distribution of Y is characterized by* $P(Y) = \sum_X P(X, Y)$ *and the conditional probability is expressed as* $P(X|Y) = \frac{P(X,Y)}{P(Y)}$. *Consequently, the definition of conditional entropy is represented as* $\mathcal{H}(X|Y) \triangleq \mathbb{E}[-\log p(X|Y)]$ *which can be further derived as* $\mathcal{H}(X|Y) = \mathbb{E}[-\log P(X, Y)] + \mathbb{E}[\log P(Y)] = \mathcal{H}(X, Y) - \mathcal{H}(Y)$.

**Definition 3 (Critic Conditioned State Entropy)** *Given empirical policy* $\pi \in \Pi$ *and its corresponded critic network* $Q^\pi : \mathcal{S} \times \mathcal{A} \to \mathbb{R}$, *and given state density of current empirical policy:* $\rho_\pi(\mathbf{s})$. *We define the critic conditioned entropy as* $\mathcal{H}_\pi(\mathbf{s}|Q^\pi) = \mathbb{E}_{\mathbf{s} \sim \rho_\pi(\mathbf{s})}[-\log p(\mathbf{s}|Q^\pi(\mathbf{s}, \pi(\mathbf{s})))]$.

**Definition 4 (State Marginal Matching)** *Given the target state density* $p^*(\mathbf{s})$ *and the offline initialized empirical state marginal distribution* $\rho_\pi(s)$. *We define State Marginal Matching (SMM) as: obtain the optimal policy to minimize* $D_{\mathrm{KL}}(\rho_\pi(\mathbf{s})||p^*(\mathbf{s}))$, *i.e.,* $\pi := \arg\min_\pi D_{\mathrm{KL}}(\rho_\pi(\mathbf{s})||p^*(\mathbf{s}))$.

## 4 SAMPLE EFFICIENCY REWARD AUGMENTATION (SERA)

Furthermore, we define Approximate State Marginal Matching (ASMM), *i.e.*, Definition 5, and then demonstrating its functionality, and then we propose SERA.

**Definition 5 (Approximate State Marginal Matching)** *Given a target state density* $p^*(\mathbf{s})$ *and the offline initialized empirical state marginal distribution* $\rho_\pi(s)$. *We define Approximate State Marginal Matching (Approximate SMM) as penalizing visitation* $\{\mathbf{s}\}$ *when* $\rho_\pi(\mathbf{s}) > p^*(\mathbf{s})$ *while encouraging state visitation* $\{\mathbf{s}'\}$ *when* $\rho_\pi(\mathbf{s}') < p^*(\mathbf{s}')$ *by maximizing state entropy,* i.e., $\pi := \arg\max_\pi \mathbb{E}_{\mathbf{s} \sim \rho_\pi(\mathbf{s})}[\mathcal{H}_\pi[\mathbf{s}]]$.

**Analysis of Definition 5** Approximate SMM can provide an approximate implementation of SMM (State Marginal Matching). *proof* see Appendix B.1. The advantage of this method lies in that it can encourage angets visiting areas where state distribution below the density,*i.e.*, $\{\mathbf{s}|\rho_\pi(\mathbf{s}) < p^*(\mathbf{s}), \mathbf{s} \in \mathcal{S}\}$ by maximizing entropy, while reducing exploration in areas of the state distribution above the density,*i.e.*, $\{\mathbf{s}|\rho_\pi(\mathbf{s}) > p^*(\mathbf{s}), \mathbf{s} \in \mathcal{S}\}$, thereby approximately realizing SMM. Continuing from **Definition 5**, we introduce SERA which can approximately realize State Marginal Matching by computing $Q$ conditioned state entropy (the advantage of Q conditioned intrinsic reward has been detailed in Section 4.3).

### 4.1 METHODOLOGY

**Reward Augmentation by SERA.** The mathematical formulation of SERA, as shown in Equation 1, involves calculating the Q-conditioned state entropy as an intrinsic reward to encourage the agent to explore the environment uniformly.

$$r^{\mathrm{mod}}(\mathbf{s}, \mathbf{a}) = \lambda \cdot \underbrace{\mathrm{Tanh}(\mathcal{H}(\mathbf{s}|\min(Q_{\phi_1}(\mathbf{s}, \mathbf{a}), Q_{\phi_2}(\mathbf{s}, \mathbf{a}))))}_{r^{\mathrm{aug}}} + r(\mathbf{s}, \mathbf{a}), \ (\mathbf{s}, \mathbf{a}) \sim \mathcal{D}_{\mathrm{online}}, \quad (1)$$

where $\phi_1$ and $\phi_2$ are separately the params of double $Q$ Networks. However, we cannot directly obtain the state density $\rho_\pi(\mathbf{s})$, therefore, we cannot directly calculate state entropy. In order to approximate $\rho_\pi(\mathbf{s})$, we refer to Kim et al. (2023), and use the KSG estimator to approximate state entropy as augmented reward, *i.e.*, Equation 2.

$$r^{\mathrm{aug}}(\mathbf{s}, \mathbf{a}) = \frac{1}{d_s}\phi(n_v(i) + 1) + \log 2 \cdot \max(||\mathbf{s}_i - \mathbf{s}_i^{knn}||, ||\hat{Q}(\mathbf{s}, \mathbf{a}) - \hat{Q}(\mathbf{s}, \mathbf{a})^{knn}||), (\mathbf{s}, \mathbf{a}) \sim \mathcal{D}_{\mathrm{online}}, \quad (2)$$

where $\hat{Q}(\mathbf{s}, \mathbf{a}) = \min(Q_{\phi_1}(\mathbf{s}, \mathbf{a}), Q_{\phi_2}(\mathbf{s}, \mathbf{a}))$, and given variable list $\{x_i\}$, $x_i^{knn}$ is the $n_x(i)$-th nearest neighbor of $x_i$. Additionally, the alternate implementation (VAE-based) of SERA has been appended into Appendix D.3.

**Training Objective.** Since SERA satisfy the guarantee of soft Q optimization, we primarily validate our method on CQL and Cal-QL, regarding the training objective of Cal-QL and CQL, we update Cal-QL's Q Network using Equation 3, and we update CQL's Q Network using Equation 4:

$$L(Q) = \mathbb{E}_{(\mathbf{s},\mathbf{a},\mathbf{s}')\sim\mathcal{D}}[(Q^\pi(\mathbf{s},\mathbf{a})-\mathcal{B}_\mathcal{M}^\pi Q(\mathbf{s},\mathbf{a}))^2]+\mathbb{E}_{\mathbf{s}\sim\mathcal{D},\mathbf{a}\sim\pi}[\max(Q^\pi(\mathbf{s},\mathbf{a}),V^\mu(\mathbf{s}))]-\mathbb{E}_{(\mathbf{s},\mathbf{a})\sim\mathcal{D}}[Q^\pi(\mathbf{s},\mathbf{a})]. \tag{3}$$

$$L(Q) = \mathbb{E}_{(\mathbf{s},\mathbf{a},\mathbf{s}')\sim\mathcal{D}}[(Q^\pi(\mathbf{s},\mathbf{a}) - \mathcal{B}_\mathcal{M}^\pi Q(\mathbf{s},\mathbf{a}))^2] + \mathbb{E}_{(\mathbf{s},\mathbf{a},\mathbf{s}')\sim\mathcal{D}}[-Q^\pi(\mathbf{s},\mathbf{a}) + Q^\pi(\mathbf{s}',\pi(\mathbf{s}'))], \tag{4}$$

where $\mathcal{D}$ is the batch training data. Meanwhile, updating their policies by Equation 5:

$$\mathcal{J}(\pi_\theta) = \mathbb{E}_{\mathbf{s}\sim\mathcal{D}}[-Q^\pi(\mathbf{s},\pi_\theta(\mathbf{s})) + \alpha\log(\pi_\theta(\mathbf{s}))]. \tag{5}$$

It's worth noting that we not only tested SERA on CQL and Cal-QL but also further extended our validation to a range of additional model-free algorithms, demonstrating the generality of SERA. These algorithms include AWAC, TD3+BC, IQL, and SAC.

## 4.2 IMPLEMENTATION OF SERA

**Implementation.** We follow the standard offline-to-online RL process to test SERA. Specifically, we first pretrain the policy with the selected algorithm on a specific offline dataset. Then, we further fine-tune the pretrained policy online using SERA. Finally, we test using the policy fine-tuned online. In terms of the real implementation, SERA augments the reward of online dataset by calculating the $Q$ conditional state entropy (via Equation 2) which is highly compatible with Q-ensemble or double-Q RL algorithms. For algorithms that do not employ Q-ensemble or double Q, it is still possible to use SERA; however, they may not benefit from the advantages associated with Q-ensemble, as clarified in the following section (theorem 4.2). When it comes to the hyper-parameters of SERA, setting $\lambda$ in Equation 1 to 1 is generally sufficient to improve the performance of various baselines on most tasks. However, it is important to note that SERA's effectiveness is influenced by the number of k-nearest neighbor (knn) clusters, as we have demonstrated in our ablation study. Additionally, for parameters unrelated to SERA, such as those of other algorithms used in conjunction with SERA, it is not necessary to adjust the original parameters of these algorithms (see more details in Appendix D.6). In the following section we will answer the following questions: 1) Can SERA guarantee policy improvement? 2) What's the advantage of Q condition over V condition?

## 4.3 ANALYSIS OF SERA

**Can SERA guarantee policy improvement?** In this section, we will provide the mathematical analysis to prove that SERA guarantee soft policy improvement. To begin with this section, we first define Soft Bellman Operator $\tau^\pi$ as Equation 6, and extend *lemma* 1 and *lemma* 2 of (Haarnoja et al., 2018) to *lemma* B.1 and *lemma* B.2 in our research and obtain our Theorem 4.1 and Theorem 4.2.

$$\tau^\pi Q(\mathbf{s}_t, \mathbf{a}_t) \triangleq \mathbb{E}_{(\mathbf{s},\mathbf{a})\sim\mathcal{D}}[Q(\mathbf{s},\mathbf{a}) - \log\pi_\beta(\cdot|\mathbf{s})], \tag{6}$$

**Theorem 4.1 (Converged SERA Soft Policy is Optimal)** *Repetitive using lemma B.1 and lemma B.2 to any $\pi \in \Pi$ leads to convergence towards a policy $\pi^*$. And it can be proved that $Q^{\pi^*}(\mathbf{s}_t, \mathbf{a}_t) \geq Q^\pi(\mathbf{s}_t, \mathbf{a}_t)$ for all policies $\pi \in \Pi$ and all state-action pairs $(\mathbf{s}_t, \mathbf{a}_t) \in \mathcal{S} \times \mathcal{A}$, provided that $|\mathcal{A}| < \infty$.*

**Theorem 4.2 (Conservative Soft Q values with SERA)** *By employing a double Q network, we ensure that in i-th iteration, the Q-value from the single Q network, denoted as $Q_{single\ Q}^{\pi_i}(\mathbf{s}_t, \mathbf{a}_t)$, is greater than or equal to the Q-value obtained from the double Q network, represented as $Q_{double\ Q}^{\pi_i}(\mathbf{s}_t, \mathbf{a}_t)$, for all $(\mathbf{s}_t, \mathbf{a}_t) \in \mathcal{S} \times \mathcal{A}$, where the action space is finite.*

See Appendix B.2 for the *proof* of *Theorem* 4.1 and *Theorem* 4.2.

Specifically, Theorem 4.1 demonstrate that when soft Q bellman operator is equipped with SERA can guarantee its monotonic policy improvement, and Theorem 4.2 demonstrate that when computing intrinsic reward with double Q network can guarantee conservative policy improvement.

**Advantages of $Q$ Condition over $V$ Condition.** A method similar to SERA is VCSE (Kim et al., 2023). The difference lies in VCSE using $V(\mathbf{s})$ calculate conditioned entropy as intrinsic rewards, and VCSE mainly focuses on pure online scenarios. The advantage of SERA is that it use pre-trained $Q$ as condition thus encouraging the agent to consider the distinctions between decisions and states while increasing exploration. (Experimental comparison has been appended top Appendix F.3)

## 5    EXPERIMENTS AND EVALUATION

The primary objectives of our experimental evaluation are as follows: **1)** We aim to investigate whether and how well SERA can facilitate offline-to-online RL. **2)** We also study the scalability of SERA on various model-free algorithms to improve their sample efficiency. **3)** Additionally, we conduct various experiments to demonstrate the performance difference or relationship between SERA and various exploration methods including SE (Seo et al., 2021), VCSE, RND (Burda et al., 2018) and SAC. **4)** Finally, we perform ablation studies to understand the feasibility of SERA. To begin with the presentation of our main results, we will first introduce our tasks and baselines.

**Task and Datasets** We experiment with 12 tasks from mujoco (Brockman et al., 2016) and Antmaze in D4RL (Fu et al., 2021). The selected tasks cover various aspects of RL challenges, including reward delay and high-dimensional continuous control. Specifically: **(1)** In the Antmaze tasks, the goal is to control a quadruped robot to reach the final goal. Notably, this agent does not receive an immediate reward for its current decision but instead only receives a reward of +1 upon successfully reaching the goal or terminating. This setup presents a form of reward delay, making these tasks adapt to evaluate the long horizontal decision-making capability of algorithms. **(2)** In Gym-locomotion tasks, the aim is to increase the agent's locomotion, which is different from Antmaze domain in that tasks of Gym-mujoco involve high-dimensional decision-making spaces. Also, the agent in Gym-mujoco has the potential to obtain rewards in real time.

**Baselines for Comparison.** For convenience, we name any algorithm **Alg** paired with SERA as **Alg**-SERA. Now we introduce our baselines. We primarily compare CQL-SERA and Cal-QL-SERA to **CQL** (Kumar et al., 2020a) and **Cal-QL** (Nakamoto et al., 2023). We also verify that SERA can be broadly plugged into various model-free algorithms including **SAC** (Haarnoja et al., 2018), **IQL** (Kostrikov et al., 2021), **TD3+BC** (Fujimoto & Gu, 2021), and **AWAC** (Nair et al., 2021), thus improving their online fine-tuning performance. In particular, Cal-QL is the recent state-of-the-art (SOTA) offline-to-online RL algorithm that has been adequately compared to multiple offline-to-online methods (O3F (Mark et al., 2023), ODT (Zheng et al., 2022), and mentioned baselines), and demonstrated obvious advantages.

### 5.1    MAIN RESULTS

We first present the results of the comparison between CQL-SERA, Cal-QL-SERA, CQL, and Cal-QL, including the online fine-tuning training curves shown in Figure 2, as well as the results after online fine-tuning displayed in Table 2. We then extend our comparison to more **Alg**-SERA and **Alg** in Figure 4. Finally, we analyze the performance differences and relationships between SERA and other exploration methods, as illustrated in Figure 5.

**Can SERA improve offline-to-online RL?** As shown in Figure. 2, SERA can improve the online fine-tuning sample efficiency, characterized by faster convergence rates. We can also observe from Table 1 that SERA maintains the online fine-tuning asymptotic performance for both CQL and Cal-QL with CQL-SERA and Cal-QL-SERA achieving the best fine-tuning results on all selected tasks). Specifically, when considering the performance after online fine-tuning, SERA yields an average improvement of 8.9% for CQL and 11.8% for Cal-QL (If we consider medium-replay, SERA can bring a 21% improvement for CQL and a 11.2% improvement for Cal-QL.), thus improving the online fine-tuning performance, additionally, we also provide statistical analysis to prove that the enhancements brought about by our approach are significant (Figure 3). It's worth noting that, CQL-SERA performs better than Cal-QL-SERA and Cal-QL on average on all tasks, which not only reflects the advantages of SERA in offline-to-online RL but also supports our view that offline-to-online performance can be improved solely from the perspective of encouraging agent's exploration.

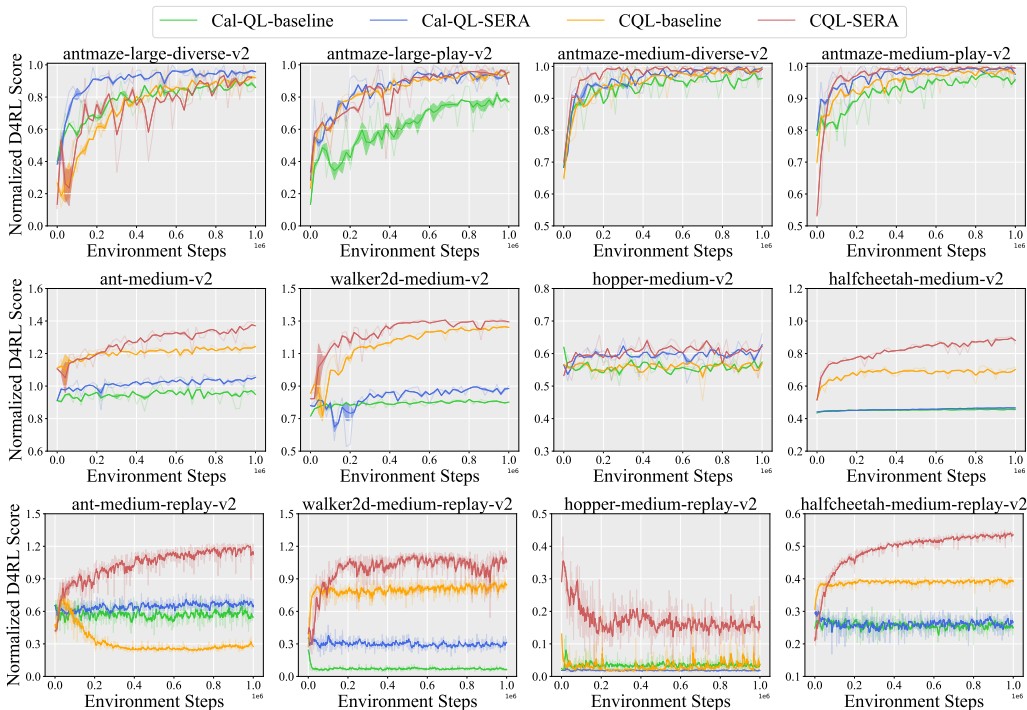

Figure 2: Online fine-tuning curve on 16 selected tasks. We tested SERA by comparing Cal-QL-SERA, CQL-SERA to Cal-QL, CQL on selected tasks in the Gym-mujoco and Antmaze domains, and then reported the average return curves of multi-time evaluation. As shown in this Figure, SERA can improve Cal-QL and CQL's offline fine-tuning sample efficiency and achieves better performance than baseline (CQL and Cal-QL *without* SERA) *over all selected tasks*.

| Task | IQL | AWAC | TD3+BC | CQL | **CQL+SERA** | Cal-QL | **Cal-QL+SERA** |
|---|---|---|---|---|---|---|---|
| antmaze-large-diverse | 59 | 00 | 00 | 89.2 | 89.8±3.2 | 86.3±0.2 | **94.5±1.7** |
| antmaze-large-play | 51 | 00 | 00 | 91.7 | 92.6± 1.3 | 83.3±9.0 | **95.0±1.1** |
| antmaze-medium-diverse | 92 | 00 | 00 | 89.6 | 98.9±0.2 | 96.8±1.0 | **99.6±0.1** |
| antmaze-medium-play | 94 | 00 | 00 | 97.7 | **99.4±0.4** | 95.8±0.9 | 98.9±0.6 |
| halfcheetah-medium | 57 | 67 | 49 | 69.9 | **87.9±2.3** | 45.6±0.0 | 46.9±0.0 |
| walker2d-meidum | 93 | 91 | 82 | 123.1 | **130.0±0.0** | 80.3±0.4 | 90.0±3.6 |
| hopper-medium | 67 | 101 | 55 | 56.4 | **62.4± 1.3** | 55.8±0.7 | 61.7±2.6 |
| ant-medium | 113 | 121 | 43 | 123.8 | **136.9±1.6** | 96.4±0.3 | 104.2±3.0 |
| **Average Fine-tuned** | 78.2 | 47.5 | 28.6 | 92.7 | **94.7** | 78.8 | 86.4 |

Table 1: Normalized score after online fine-tuning. We report the online fine-tuned normalized return. SERA obviously improves the performance of CQL and Cal-QL. In particular, CQL-SERA (mean score of **94.7**) is the best out of the 8 selected baselines. Notably, part of Antmaze's baseline results are *quoted* from existing studies. Among them, AWAC's results are *quoted* from Kostrikov et al. (2021) and CQL's results are *quoted* from Nakamoto et al. (2023).

**Can SERA be plugged into various model-free algorithms?** To answer the second question, we conduct comparative experiments to test our SERA on various model-free algorithms including TD3+BC, AWAC, IQL, SAC. Importantly, since our SERA is a plugged reward augmentation algorithm, it does not require any additional modifications (*i.e.*, we simply incorporate SERA to modify the reward when training on those algorithms). As shown in Figure.4, when SERA is plugged in, almost all algorithms gain performance improvements during online fine-tuning, showing that SERA can be applied effectively to a wide range of RL algorithms beyond the scope of CQL or Cal-QL.

**SERA and various Exploration methods in offline-to-online.** In this section, we compare and demonstrate the performance difference between SERA and several related exploration methods including VCSE, SE and RND, and we also explored the changes in performance when SAC is

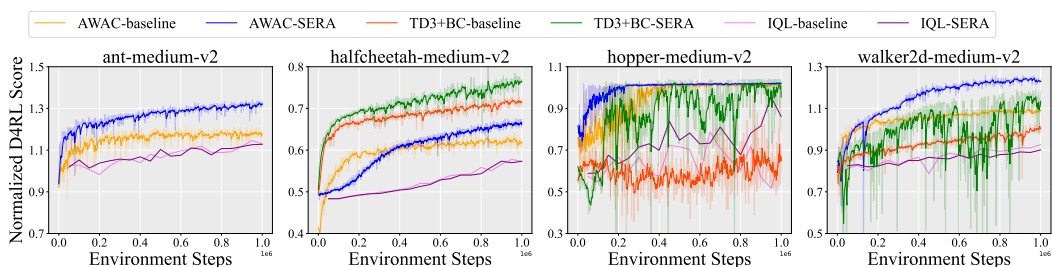

Figure 3: Aggregate metrics with SERA. We refer to Agarwal et al. (2022) to conduct the statistical analysis of SERA. Specifically, higher median, IQM, and mean scores are better, SERA can significantly improve the performance of CQL and Cal-QL.

Figure 4: Performance of **Alg**-SERA. We test SERA with AWAC, TD3+BC, and IQL on selected Gym-mujoco tasks, SERA can obviously improve the performance of these algorithms on selected Gym-mujoco tasks, showing SERA's versility.

combined with SERA. As shown in Figure 5 (a), when SAC is combined with SERA, it can enhance the performance of SAC on the selected gym-mujoco tasks. This experimental result is consistent with Theorem 4.1, which states that SERA can ensure the monotonic soft Q optimization. As shown in Figure 5 (b), we compared the experimental results of IQL and AWAC with different reward augmentation methods (SEAR, RND, and SE), and we found that increasing exploration can improve the performance of both IQL and AWAC. Moreover, algorithms combined with SERA perform the best on all selected tasks and are overall more stable. This further proves that SERA ensures the monotonic soft Q optimization and highlights the advantage of Q condition.

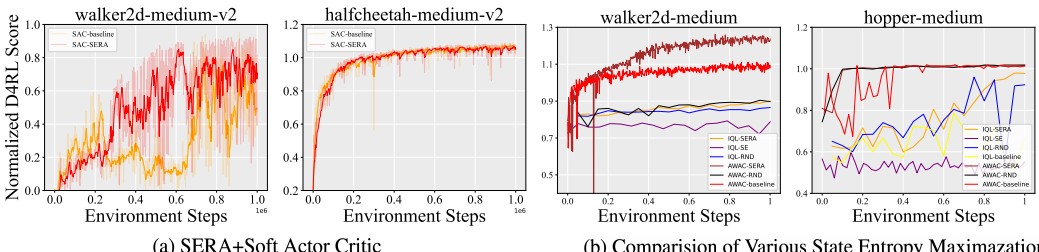

(a) SERA+Soft Actor Critic

(b) Comparision of Various State Entropy Maximazation

Figure 5: Performance comparison for variety exploration Methods. (a) Online fine-tuning performance difference between SAC and SAC-SERA. (b) Online fine-tuning performance difference between SERA, VCSE and SE with IQL. SERA performs the best over selected algorithms.

## 5.2 ABLATIONS

**Effect of Hyperparameter.** We now focus on quantifying the impact of SERA on the performance of online finetuning. Thus we mainly study the effect of Equation.19's hyperparameter, as shown in section 4. The state entropy is approximated via KSG estimator, where the number of state clusters serves as a crucial hyperparameter. As shown in Figure.2, the performance can indeed be influenced by the number of state clusters, and a trade-off exists among the sizes of these state clusters. For instance, the optimal cluster settings of walker2d and hopper are saturated around 20 and 10, respectively. In contrast, a task like antmaze-large-diverse requires a larger number of clusters (*about* 25). We consider the main reason is that different tasks require varying degrees of exploration, and thus need different cluster settings. Therefore, our SERA proves to be valid.

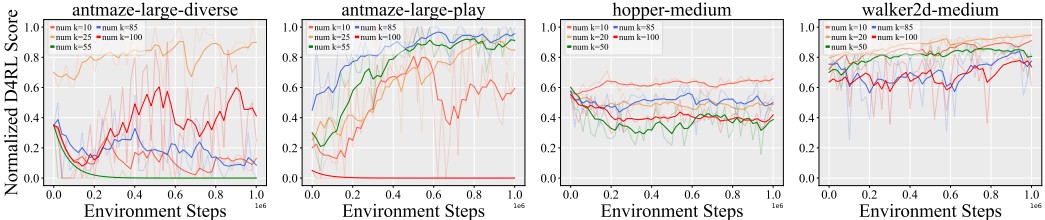

Figure 6: We evaluate the performance difference that arises when varying the number of state clusters. We assess SERA by configuring different sizes of k-nearest neighbor (knn) clusters and subsequently observe the impact of these parameter settings on online fine-tuning, and it can be observed that the choice of knn cluster settings exerts a notable influence on SERA's performance.

**SERA vs. Various Efficient Algorithms** In order to more intuitively demonstrate the effectiveness of SERA, we replaced SERA with a series of past efficient offline-to-online algorithms and conducted comparisons. As shown in Figure 7, we select CQL as the base algorithm and aggregate it with SERA, APL (Zheng et al., 2023), PEX (Zhang et al., 2023) and BR (Lee et al., 2021a) to test on tasks of Antmaze and Gym-mujoco (`medium`, `medium-replay`) domains, and CQL-SERA archives the best performance (**83.8**) over all selected baselines, which demonstrating SERA has better performance than previous efficient offline-to-online algorithms. (Experimental results of Figure 7 has been appended to Table 10)

**Extended Ablations.** We conduct additional ablation experiments to validate the effectiveness of SERA, and we chose AWAC as the test target. Specifically, we compared the effects of using an offline pre-trained Q-network and a randomly initialized Q-network to compute rewards in Figure 8 (a). Offline pre-training of the Q-network leads to improved algorithm performance, while training the Q-network from scratch results in the model's performance falling below the baseline. In Figure 8 (b), we visualize the change in state entropy as training pro-

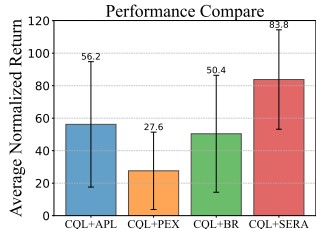

Figure 7: Performance Comparison.

gresses. Specifically, we observe that the state entropy of AWAC combined with SERA eventually surpasses the state entropy of the baseline, which demonstrates that SERA influences state entropy.

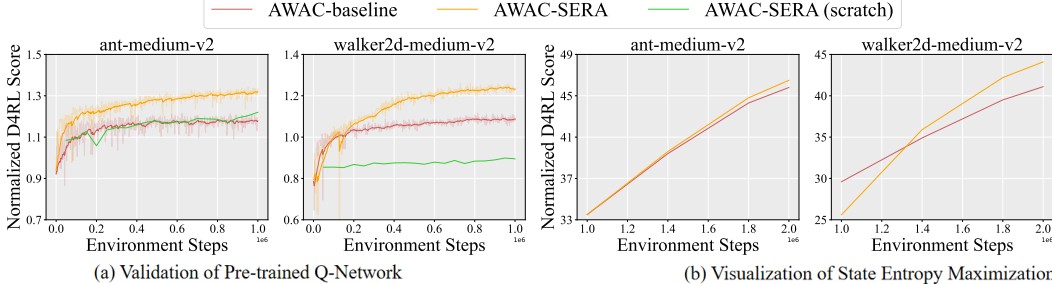

Figure 8: (a) Ablation experiments to validate the impact of pre-trained Q network. (b) Quantitative results on the agent's state entropy. From Figure (a), we can deduce that SERA . According to the results from Figure (b), we can deduce that SERA increases the agent's state entropy, which aligns with theoretical expectations.

## 6 CONCLUSIONS

In this study, we proposed a general offline-to-online framework called SERA. On a theoretical level, we demonstrated that SERA ensures the optimization of algorithms based on soft Q. On an experimental level, SERA led to improvements for both CQL and Cal-QL, validating our theoretical claims. We also extend the test of SERA to other model-free algorithms, and experimental results showed that SERA performs well when combined with other model-free algorithms, demonstrating its generality. Additionally, we conduct extensive ablations and compared SERA with a series of previous efficient offline-to-online algorithms (APL, PEX, etc.) in terms of performance, and we found that SERA outperforms the majority of these efficient offline-to-online algorithms.

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

CONTENTS

**G Extended Related Work**      **26**

# A   ETHICAL CLAIM

Despite the potential of offline RL to learn from the static datasets without the necessity to access the online environment, the offline method does not guarantee the optimal policy. Therefore, online fine-tuning is essential for policy improvement. In this study, we propose a novel and versatile reward augmentation framework, named Sample Efficient Reward Augmentation (SERA) which can be seamlessly plugged into various model-free algorithms. We believe our approach is constructive and will enhance the sample efficiency of offline-to-online RL. Additionally, given that SERA is an integrated algorithm, we also believe it can broadly and readily benefit existing algorithms.

# B   THEORETICAL ANALYSIS

In this section, we provide the supplementary mathematical analysis for SERA.

## B.1   ANALYSIS OF APPROXIMATE STATE MARGINAL MATCHING.

**Approximate Marginal Matching (ASMM).**   Given the empirical state distribution $\rho_\pi(\mathbf{s})$ under current empirical policy $\pi$ and target density $p^*(\mathbf{s})$, the optimization of $\min D_{\mathrm{KL}}(\rho_\pi(\mathbf{s})||p^*(\mathbf{s}))$ is equivalent to Equation 7.

$$\min D_{\mathrm{KL}}(\rho_\pi(\mathbf{s})||p^*(\mathbf{s})) \triangleq \max \mathbb{E}_{\mathbf{s}\sim\rho_\pi(\mathbf{s})}[\log p^*(\mathbf{s}) + \mathcal{H}_\pi[\mathbf{s}]]. \tag{7}$$

**ASMM.**   In section 3.2, due to the absence of an explicit definition for $p^*$, we propose the concept of implicit SMM, *i.e.*, $\min D_{\mathrm{KL}}(\rho_\pi(\mathbf{s})||p^*(\mathbf{s})) \approx \max \mathbb{E}_{\mathbf{s}\sim\rho_\pi(\mathbf{s})}[\mathcal{H}_\pi[\mathbf{s}]]$. Here we will provide a complementary analysis to assess the feasibility of this method.

**Why does ASMM encourage covering the target density?**   We commence our analysis by expressing $\max \mathbb{E}_{\mathbf{s}\sim\rho_\pi(\mathbf{s})}[\mathcal{H}_\pi(\mathbf{s})]$ in an alternative form:

$$\begin{aligned}
\max \mathbb{E}_{\mathbf{s}\sim\rho_\pi(\mathbf{s})}[\mathcal{H}_\pi(\mathbf{s})] &= \max \mathbb{E}_{\mathbf{s}\sim\rho_\pi(\mathbf{s})}[-\log \rho_\pi(s)] \\
&= \max \int_{\mathbf{s}\sim dom(\rho_\pi)} -\rho_\pi(\mathbf{s})\log \rho_\pi(\mathbf{s})d\mathbf{s} \\
&= \max \int_{\mathbf{s}\sim dom(\rho_\pi)} -\rho_\pi(\mathbf{s})\log(\frac{\rho_\pi(\mathbf{s})}{p^*(\mathbf{s})} \times p^*(\mathbf{s}))d\mathbf{s} \\
&= \max \int_{\mathbf{s}\sim dom(\rho_\pi)} -\rho_\pi(\mathbf{s})(\log p^*(\mathbf{s}) + \log \frac{\rho_\pi(\mathbf{s})}{p^*(\mathbf{s})})d\mathbf{s} \\
&= \min \int_{\mathbf{s}\sim dom(\rho_\pi)} \rho_\pi(\mathbf{s})\log p^*(\mathbf{s}) + \rho_\pi(\mathbf{s})\log \frac{\rho_\pi(\mathbf{s})}{p^*(\mathbf{s})}d\mathbf{s}
\end{aligned}$$

where $dom(\rho_\pi)$ donates the domain of state space under function $\rho_\pi$. Subsequently, we employ a logarithmic inequality, i.e. $1 - \frac{1}{x} \le \log x \le x - 1$, to further derive the aforementioned expression:

$$\begin{aligned}
\min \int_{\mathbf{s}\sim dom(\rho_\pi)} \rho_\pi(\mathbf{s})\log p^*(\mathbf{s}) + \rho_\pi(\mathbf{s})\log \frac{\rho_\pi(\mathbf{s})}{p^*(\mathbf{s})}d\mathbf{s} &\ge \min \int_{\mathbf{s}\sim dom(\rho_\pi)} \rho_\pi(\mathbf{s}) - \frac{\rho_\pi(\mathbf{s})}{p^*(\mathbf{s})} + \rho_\pi(\mathbf{s}) - p^*(\mathbf{s})d\mathbf{s} \\
&= 2 - \max \int_{\mathbf{s}\sim dom(\rho_\pi(\mathbf{s}))} \frac{\rho_\pi(\mathbf{s})}{p^*(\mathbf{s})} + p^*(\mathbf{s})d\mathbf{s} \\
&\ge 2 - \max \int_{\mathbf{s}\sim dom(\rho_\pi(\mathbf{s}))} \frac{1}{p^*(\mathbf{s})} + p^*(\mathbf{s})d\mathbf{s}
\end{aligned}$$

It is worth noting that, given that $p^*(\mathbf{s})$ is the fixed target state density and $p^*(\mathbf{s}) \in [0, 1]$ for all $\mathbf{s} \in dom(p^*)$, we have $(\frac{1}{p^*(\mathbf{s})} + p^*(\mathbf{s})) > 0$. Therefore, the process of maximising $\mathbb{E}_{\mathbf{s}\sim\rho_\pi(\mathbf{s})}[\mathcal{H}[\mathbf{s}]]$ is equivalent to maximizing $\int_{\mathbf{s}\sim dom(\rho_\pi(\mathbf{s}))} \frac{1}{p^*(\mathbf{s})} + p^*(\mathbf{s})d\mathbf{s}$. This leads the domain of $\rho_\pi(\mathbf{s})$(which is initially smaller than the domain of $p^*(\mathbf{s})$ due to limited state exploration at the beginning of state entropy maximization) to cover the domain of $p^*(\mathbf{s})$.

**Trade off between $\rho_\pi$ and $p^*$ when maximizing entropy.** We further derivate Equation B.1 and obtained :

$$\max \mathbb{E}_{\mathbf{s}\sim\rho_\pi(\mathbf{s})}[\mathcal{H}_\pi(\mathbf{s})] = \max \mathbb{E}_{\mathbf{s}\sim\rho_\pi(\mathbf{s})}[-\log \rho_\pi(s)]$$

$$= \max \int_{\mathbf{s}\sim dom(\rho_\pi)} -\rho_\pi(\mathbf{s}) \log \rho_\pi(\mathbf{s}) d\mathbf{s}$$

$$= \max \int_{\mathbf{s}\sim dom(\rho_\pi)} -\rho_\pi(\mathbf{s}) \log(\frac{\rho_\pi(\mathbf{s})}{p^*(\mathbf{s})} \times p^*(\mathbf{s})) d\mathbf{s}$$

$$= \max \int_{\mathbf{s}\sim dom(\rho_\pi)} -\rho_\pi(\mathbf{s}) \log p^*(\mathbf{s}) - \rho_\pi(\mathbf{s}) \log \frac{\rho_\pi(\mathbf{s})}{p^*(\mathbf{s})} d\mathbf{s}$$

$$= \min \underbrace{\int_{\mathbf{s}\sim dom(\rho_\pi)} \rho_\pi(\mathbf{s}) \log p^*(\mathbf{s}) \, d\mathbf{s}}_{\text{term.1}} + \max \underbrace{\int_{\mathbf{s}\sim dom(\rho_\pi)} -\rho_\pi(\mathbf{s}) \log \frac{\rho_\pi(\mathbf{s})}{p^*(\mathbf{s})} \, d\mathbf{s}}_{\text{term.2}}$$

*Analysis term.1:* We further derive $\text{term}_1$:

At first,

$$\mathcal{J}_{\text{term}_1} = \max \int_{\mathbf{s}\sim dom(\rho_\pi)} \rho_\pi(\mathbf{s}) \log \frac{1}{p^*(\mathbf{s})} d\mathbf{s}$$

$$\geq \int_{\mathbf{s}\sim dom(\rho_\pi)} \rho_\pi(\mathbf{s}) \log \frac{\rho_\pi(\mathbf{s})}{p^*(\mathbf{s})} d\mathbf{s} \tag{8}$$

$$= D_{\text{KL}}(\rho_\pi(\mathbf{s})||p^*(\mathbf{s}))$$

meanwhile, we study $\mathcal{J}_{\text{term}_1} - D_{\text{KL}}(\rho_\pi(\mathbf{s})||p^*(\mathbf{s}))$.

$$\mathcal{J}_{\text{term}_1} - D_{\text{KL}}(\rho_\pi(\mathbf{s})||p^*(\mathbf{s}))$$

$$= \int_{\mathbf{s}\sim dom(\rho_\pi)} \rho_\pi(\mathbf{s}) \log \frac{1}{p^*(\mathbf{s})} - \rho_\pi \log \frac{\rho_\pi(\mathbf{s})}{p^*(\mathbf{s})} d\mathbf{s}$$

$$= \int_{\mathbf{s}\sim dom(\rho_\pi)} \rho_\pi(\mathbf{s}) \log \frac{1}{\rho_\pi(\mathbf{s})} d\mathbf{s}$$

$$\leq \int_{\mathbf{s}\sim dom(\rho_\pi)} \rho_\pi(\mathbf{s})[\frac{1}{\rho_\pi(\mathbf{s})} - 1] d\mathbf{s} \tag{9}$$

$$= \int_{\mathbf{s}\sim dom(\rho_\pi)} (1 - \rho_\pi(\mathbf{s})) d\mathbf{s}$$

$$\leq \int_{\mathbf{s}\sim dom(\rho_\pi)} d\mathbf{s}$$

Therefore, $\mathcal{J}_{\text{term}_1} \leq (D_{\text{KL}}(\rho_\pi(\mathbf{s})||p^*(\mathbf{s})) + \int_{\mathbf{s}\sim dom(\rho_\pi)} d\mathbf{s})$ and minimizing $\text{term}_1$ is equivalent to maximizing the KL divergence between $\rho_\pi(\mathbf{s})$ and $p^*(\mathbf{s})$, then push $\rho_\pi(\mathbf{s})$ away from $p^*(\mathbf{s})$.

*Analysis term.2:* We can observe that term.2 is a form of KL deivergence:

$$\mathcal{J}_{\text{term}_2} = \max \int_{\mathbf{s}\sim dom(\rho_\pi)} -\rho_\pi(\mathbf{s}) \log \frac{\rho_\pi(\mathbf{s})}{p^*(\mathbf{s})}$$

$$= \min D_{\text{KL}}(\rho_\pi(\mathbf{s})||p^*(\mathbf{s})), \tag{10}$$

Thus, optimizeing term.2 is equiv to minimize the KL divergence between $p^*(\mathbf{s})$ and $\rho_\pi(\mathbf{s})$, thereby driving $\rho_\pi$ approaching $p^*(\mathbf{s})$.

*Analysis Summary:* In conclusion, based on the Analysis term.1 and Analysis term.2, it can be deduced that the optimization of term 1 makes $\rho_\pi(\mathbf{s})$ away from $p^*(\mathbf{s})$, whereas the optimization of Term 2 facilitates the convergence of $\rho_\pi(\mathbf{s})$ towards $p^*(\mathbf{s})$. Therefore, these two objectives represent a trade-off, offering the advantage of encouraging the agent to approach the target distribution while maintaining its capacity for exploration.

### B.2 MATHEMATICS ANALYSIS OF SERA ALGORITHM

In this section, we examine the mathematical viability of the SERA framework, focusing on two key aspects: **1)** Guarantee of Soft policy optimization **2)** Prevention of OOD state actions.

We first introduce the modified soft Q Bellman backup operator, denoted as Equation 11,

$$\mathcal{T}_{sera}^{\pi} Q\left(\mathbf{s}_t, \mathbf{a}_t\right) \triangleq r\left(\mathbf{s}_t, \mathbf{a}_t\right) + r^{aug}\left(\mathbf{s}_t, \mathbf{a}_t\right) + \gamma \mathbb{E}_{\mathbf{s}_{t+1} \sim p}\left[V\left(\mathbf{s}_{t+1}\right)\right] \tag{11}$$

In this equation, the term $V\left(\mathbf{s}_t\right) = \mathbb{E}_{\mathbf{a}_t \sim \pi}\left[Q\left(\mathbf{s}_t, \mathbf{a}_t\right) - \log \pi\left(\mathbf{a}_t \mid \mathbf{s}_t\right)\right]$ is defined.

**Lemma B.1 (Soft Policy Evaluation with SERA.)** *Given the modified soft bellman backup operator $\mathcal{T}_{sera}^{\pi}$ in Equation 11, along with a mapping $Q^0 : \mathcal{S} \times \mathcal{A} \to \mathbb{R}$ where $|\mathcal{A}| < \infty$. We define an iterative sequence as $Q^{k+1} = \mathcal{T}^{\pi} Q^k$. It can be shown that when index $k$ tends towards infinity, the sequence $Q^k$ converges to a soft Q-value of $\pi$.*

*proof.* Let us define the SERA reward as follows

$$r_{sera}^{\pi}\left(\mathbf{s}_t, \mathbf{a}_t\right) \triangleq r\left(\mathbf{s}_t, \mathbf{a}_t\right) + \lambda \operatorname{Tanh}\left(\mathcal{H}\left(\mathbf{s}_t \mid \min\left(Q_{\phi_1}\left(\mathbf{s}_t, \mathbf{a}_t\right), Q_{\phi_2}\left(\mathbf{s}_t, \mathbf{a}_t\right)\right)\right)\right) + \mathbb{E}_{\mathbf{s}_{t+1} \sim p}\left[\mathcal{H}\left(\pi\left(\cdot \mid \mathbf{s}_{t+1}\right)\right)\right] \tag{12}$$

and rewrite the update rule as

$$Q\left(\mathbf{s}_t, \mathbf{a}_t\right) \leftarrow r_{sera}^{\pi}\left(\mathbf{s}_t, \mathbf{a}_t\right) + \gamma \mathbb{E}_{\mathbf{s}_{t+1} \sim p, \mathbf{a}_{t+1} \sim \pi}\left[Q\left(\mathbf{s}_{t+1}, \mathbf{a}_{t+1}\right)\right]. \tag{13}$$

Then we can apply mathematical analysis of convergence for policy evaluation as outlined in Sutton & Barto (1998) to prove the result. It is essential to note that the assumption $|\mathcal{A}| < \infty$ is necessary to ensure the boundedness of the SERA reward."

**Lemma B.2 (Soft Policy Improvement with SERA)** *Let $\pi_{old} \in \Pi$, and let $\pi_{new}$ be the solution to the minimization problem defined as:*

$$\pi_{\text{new}} = \arg \min_{\pi' \in \Pi} \mathrm{D}_{\mathrm{KL}}\left(\pi'\left(\cdot \mid \mathbf{s}_t\right) \| \frac{\exp\left(Q^{\pi_{\text{old}}}\left(\mathbf{s}_t, \cdot\right)\right)}{Z^{\pi_{\text{old}}}\left(\mathbf{s}_t\right)}\right). \tag{14}$$

*Then, it follows that $Q^{\pi_{new}}\left(\mathbf{s}_t, \mathbf{a}_t\right) \geq Q^{\pi_{old}}\left(\mathbf{s}_t, \mathbf{a}_t\right)$ for all $\left(\mathbf{s}_t, \mathbf{a}_t\right) \in \mathcal{S} \times \mathcal{A}$ provided that $|\mathcal{A}| < \infty$.*

*proof.* Starting from Equation 15, which has been established in the work by (Haarnoja et al., 2018), as:

$$\mathbb{E}_{\mathbf{a}_t \sim \pi_{\text{new}}}\left[Q^{\pi_{\text{old}}}\left(\mathbf{s}_t, \mathbf{a}_t\right) - \log \pi_{\text{new}}\left(\mathbf{a}_t \mid \mathbf{s}_t\right)\right] \geq V^{\pi_{\text{old}}}\left(\mathbf{s}_t\right), \tag{15}$$

we proceed to consider the soft Bellman equation, which can be expressed as:

$$
\begin{aligned}
Q^{\pi_{\text{old}}}\left(\mathbf{s}_t, \mathbf{a}_t\right) &= r\left(\mathbf{s}_t, \mathbf{a}_t\right) + r_{aug}\left(\mathbf{s}_t, \mathbf{a}_t\right) + \gamma \mathbb{E}_{\mathbf{s}_{t+1} \sim p}\left[V^{\pi_{\text{old}}}\left(\mathbf{s}_{t+1}\right)\right] \\
&\leq r\left(\mathbf{s}_t, \mathbf{a}_t\right) + r_{aug}\left(\mathbf{s}_t, \mathbf{a}_t\right) + \gamma \mathbb{E}_{\mathbf{s}_{t+1} \sim p}\left[\mathbb{E}_{\mathbf{a}_{t+1} \sim \pi_{\text{new}}}\left[Q^{\pi_{\text{old}}}\left(\mathbf{s}_{t+1}, \mathbf{a}_{t+1}\right) - \log \pi_{\text{new}}\left(\mathbf{a}_{t+1} \mid \mathbf{s}_{t+1}\right)\right]\right] \\
&\vdots \\
&\leq Q^{\pi_{\text{new}}}\left(\mathbf{s}_t, \mathbf{a}_t\right)
\end{aligned}
\tag{16}
$$

Here, we have iteratively expanded $Q^{\pi_{\text{old}}}$ on the right-hand side by applying both the soft Bellman equation and the inequality from Equation 15.

**Theorem B.3 (Converged SERA Soft Policy is Optimal)** *Repetitive using Lemma 1 and Lemma 2 to any $\pi \in \Pi$ leads to convergence towards a policy $\pi^*$. And it can be proved that $Q^{\pi^*}\left(\mathbf{s}_t, \mathbf{a}_t\right) \geq Q^{\pi}\left(\mathbf{s}_t, \mathbf{a}_t\right)$ for all policies $\pi \in \Pi$ and all state-action pairs $\left(\mathbf{s}_t, \mathbf{a}_t\right) \in \mathcal{S} \times \mathcal{A}$, provided that $|\mathcal{A}| < \infty$.*

*proof.*

Let $\pi_i$ represent the policy at iteration $i$. According to Lemma 2, the sequence $Q^{\pi_i}$ exhibits a monotonic increase. Given that rewards and entropy and thus $Q^{\pi}$ are bounded from above for policies within the set $\Pi$, the sequence converges to a certain policy $\pi^*$. It is essential to demonstrate that $\pi^*$ is indeed an optimal policy. Utilizing a similar iterative argument as employed in the proof of Lemma 2, we can establish that $Q^{\pi^*}\left(\mathbf{s}_t, \mathbf{a}_t\right) > Q^{\pi}\left(\mathbf{s}_t, \mathbf{a}_t\right)$ holds for all $\left(\mathbf{s}_t, \mathbf{a}_t\right) \in \mathcal{S} \times \mathcal{A}$. In other words, the soft value associated with any other policy in $\Pi$ is lower than that of the converged policy. Consequently, $\pi^*$ is confirmed as the optimal policy within the set $\Pi$.

**Theorem B.4 (Conservative Soft Q values with SERA)** *By employing a double Q network, we ensure that in each iteration, the Q-value from the single Q network, denoted as $Q_{single\ Q}^{\pi_i}(\mathbf{s}_t, \mathbf{a}_t)$, is greater than or equal to the Q-value obtained from the double Q network, represented as $Q_{double\ Q}^{\pi_i}(\mathbf{s}_t, \mathbf{a}_t)$, for all $(\mathbf{s}_t, \mathbf{a}_t) \in \mathcal{S} \times \mathcal{A}$, where the action space is finite.*

*proof.* Let's begin by defining $\hat{Q}(\mathbf{s}_t, \mathbf{a}_t) = \min(Q_{\phi_1}(\mathbf{s}_t, \mathbf{a}_t), Q_{\phi_2}(\mathbf{s}_t, \mathbf{a}_t))$. We then proceed to examine the difference between the augmented rewards in the context of SERA for the single Q and double Q networks:

$$r_{aug}(\mathbf{s}_t, \mathbf{a}_t | \hat{Q}(\mathbf{s}_t, \mathbf{a}_t)) - r_{aug}(\mathbf{s}_t, \mathbf{a}_t | Q(\mathbf{s}_t, \mathbf{a}_t))$$

$$= \sum_{i=0}^{N} \log 2 \max(||s_i - s_i^{knn}||, ||\hat{Q}(\mathbf{s}_t, \mathbf{a}_t) - \hat{Q}^{knn}(\mathbf{s}_t, \mathbf{a}_t)||) -$$

$$\sum_{i=0}^{N} \log 2 \max(||s_i - s_i^{knn}||, ||Q(\mathbf{s}_t, \mathbf{a}_t) - Q^{knn}(\mathbf{s}_t, \mathbf{a}_t)||)$$

$$= \log \frac{\prod_{i=0}^{N} \max(||s_i - s_i^{knn}||, ||\hat{Q}(\mathbf{s}_t, \mathbf{a}_t) - \hat{Q}^{knn}(\mathbf{s}_t, \mathbf{a}_t)||)}{\prod_{i=0}^{N} \max(||s_i - s_i^{knn}||, ||Q(\mathbf{s}_t, \mathbf{a}_t) - Q^{knn}(\mathbf{s}_t, \mathbf{a}_t))||}$$

$$\approx \log \frac{\prod_{i=0}^{N} \max(||s_i - s_i^{knn}||, \mathcal{H}(\hat{Q}))}{\prod_{i=0}^{N} \max(||s_i - s_i^{knn}||, \mathcal{H}(Q)||)}$$

$$\leq \log \frac{\prod_{i=0}^{N} \max(||s_i - s_i^{knn}||, \mathcal{H}(Q))}{\prod_{i=0}^{N} \max(||s_i - s_i^{knn}||, \mathcal{H}(Q))} = 0$$

$$(17)$$

Consequently, we establish that $r_{aug}(\mathbf{s}_t, \mathbf{a}_t | \hat{Q}(\mathbf{s}_t, \mathbf{a}_t)) \leq r_{aug}(\mathbf{s}_t, \mathbf{a}_t | Q(\mathbf{s}_t, \mathbf{a}_t))$. Now we consider the modified soft Bellman equation

$$Q_{double\ Q}^{\pi_i}(\mathbf{s}_t, \mathbf{a}_t)$$

$$= r(\mathbf{s}_t, \mathbf{a}_t) + r_{aug}(\mathbf{s}_t, \mathbf{a}_t | \hat{Q}(\mathbf{s}_t, \mathbf{a}_t)) + \gamma \cdot \mathbb{E}_{\mathbf{s}_{t+1} \sim p}[\hat{V}(\mathbf{s}_{t+1})]$$

$$= r(\mathbf{s}_t, \mathbf{a}_t) + r_{aug}(\mathbf{s}_t, \mathbf{a}_t | \hat{Q}(\mathbf{s}_t, \mathbf{a}_t)) + \gamma \cdot \mathbb{E}_{\mathbf{s}_{t+1} \sim p, \mathbf{a}_{t+1} \sim \pi} \left[ \hat{Q}(\mathbf{s}_{t+1}, \mathbf{a}_{t+1}) - \log \pi(\mathbf{a}_{t+1} \mid \mathbf{s}_{t+1}) \right]$$

$$\vdots$$

$$= r(\mathbf{s}_t, \mathbf{a}_t) + r_{aug}(\mathbf{s}_t, \mathbf{a}_t | \hat{Q}(\mathbf{s}_t, \mathbf{a}_t)) + \gamma \cdot \mathbb{E}_{\mathbf{s}_{t+1} \sim p, \mathbf{a}_{t+1} \sim \pi}[r^{mod}(\mathbf{s}_{t+1}, \mathbf{a}_{t+1} | \hat{Q}(\mathbf{s}_{t+1}, \mathbf{a}_{t+1}))] \cdots +$$

$$\gamma^n \cdot \mathbb{E}_{\mathbf{s}_{t+n} \sim p, \mathbf{a}_{t+n} \sim \pi}[r^{mod}(\mathbf{s}_{t+n}, \mathbf{a}_{t+n} | \hat{Q}(\mathbf{s}_{t+n}, \mathbf{a}_{t+n}))] + \cdots + \text{entropy terms}$$

$$\leq r(\mathbf{s}_t, \mathbf{a}_t) + r_{aug}(\mathbf{s}_t, \mathbf{a}_t | Q(\mathbf{s}_t, \mathbf{a}_t)) + \gamma \cdot \mathbb{E}_{\mathbf{s}_{t+1} \sim p, \mathbf{a}_{t+1} \sim \pi}[r^{mod}(\mathbf{s}_{t+1}, \mathbf{a}_{t+1} | Q(\mathbf{s}_{t+1}, \mathbf{a}_{t+1}))] \cdots +$$

$$\gamma^n \cdot \mathbb{E}_{\mathbf{s}_{t+n} \sim p, \mathbf{a}_{t+n} \sim \pi}[r^{mod}(\mathbf{s}_{t+n}, \mathbf{a}_{t+n} | Q(\mathbf{s}_{t+n}, \mathbf{a}_{t+n}))] + \cdots + \text{entropy terms}$$

$$= Q_{single\ Q}^{\pi_i}(\mathbf{s}_t, \mathbf{a}_t)$$

$$(18)$$

where we have repeatedly expanded $\hat{Q}$ in terms of SERA rewards to obtain the final inequality $Q_{singleQ}^{\pi_i} \geq Q_{double\ Q}^{\pi_i}$.

## C    EXPERIMENTAL SETUP

In this section, we introduce the benchmarks and dataset we utilized, specifically, we mainly utilize gym-mujoco and antmaze to test our algorithm.

### C.1    GYM MUJOCO

Our benchmars from gym-mujoco domain mainly includes `halfcheetah`, `ant`, `hopper` and `walker2d`, and concrete information of these benchmarks can be referred to table 2. In paticular, the action and observation space of these locomotion benchmarks are continuous and any decision making will receive an immediate reward.

| Environment | Task Name | Samples | Observation Dim | Action Dim |
|---|---|---|---|---|
| `halfcheetah` | medium | $10^6$ | 6 | 17 |
| `walker2d` | medium | $10^6$ | 6 | 17 |
| `hopper` | medium | $10^6$ | 3 | 11 |
| `ant` | medium | $10^6$ | 8 | 111 |
| `halfcheetah` | medium-replay | $2.02 \times 10^5$ | 6 | 17 |
| `walker2d` | medium-replay | $3.02 \times 10^5$ | 6 | 17 |
| `hopper` | medium-replay | $4.02 \times 10^5$ | 3 | 11 |
| `ant` | medium-replay | $3.02 \times 10^5$ | 8 | 111 |

Table 2: Introduction of D4RL tasks (Gym-Mujoco).

### C.2    ANTMAZE

Our benchmars from antmaze mainly includes `antmaze-large-diverse`, `antmaze-medium-diverse`, `antmaze-large-play` and `antmaze-medium-play`, concrete information of our benchmarks can be referred to table 3.

| Environment | Task Name | Samples | Observation Dim | Action Dim |
|---|---|---|---|---|
| `antmaze` | large-diverse | $10^6$ | 29 | 8 |
| `antmaze` | large-play | $10^6$ | 29 | 8 |
| `antmaze` | medium-diverse | $10^6$ | 29 | 8 |
| `antmaze` | medium-play | $10^6$ | 29 | 8 |

Table 3: Introduction of D4RL tasks (Antmaze).

## D    IMPLANTATION DETAILS

### D.1    OFFLINE-TO-ONLINE IMPLANTATION

The workflow of our method is similar to the most of offline-to-online algorithms that we firstly pre-train on offline datasets, followed by online fine-tuning (Interacting with online environment to collect online dataset and followed by fine-tuning on offline and online datasets).

### D.2    EVALUATION DETAILS

Our evaluation method can be refered to Fu et al. (2021). That is for each evaluation, we freeze the parameter of trained model, and then conducting evaluation $10 \sim 50$ times and then computing the normalized score via $\frac{\text{score}_{\text{evaluation}} - \text{score}_{\text{expert}}}{\text{score}_{\text{expert}} - \text{score}_{\text{random}}}$, and then averaging these normalized evaluation scores.

### D.3    SERA IMPLANTATION

In SERA framework, we modify our reward as :

$$r^{\text{mod}}(\mathbf{s}, \mathbf{a}) = \lambda \cdot \underbrace{\text{Tanh}(\mathcal{H}(\mathbf{s}|\min(Q_{\phi_1}(\mathbf{s}, \mathbf{a}), Q_{\phi_2}(\mathbf{s}, \mathbf{a}))))}_{r^{\text{aug}}} + r(\mathbf{s}, \mathbf{a}), \quad (\mathbf{s}, \mathbf{a}) \sim \mathcal{D}_{\text{online}} \qquad (19)$$

To calculate the intrinsic reward $r^{\mathrm{aug}}$ for the online replay buffer $D_{\mathrm{online}}$, we use the KSG estimator, as defined in Equation 20, to estimate the conditional state density of the empirical dataset $D_{\mathrm{online}}$

$$r^{\mathrm{aug}}(\mathbf{s}, \mathbf{a}) = \frac{1}{d_s}\phi(n_v(i)+1)+\log 2\cdot\max(||\mathbf{s}_i-\mathbf{s}_i^{knn}||,||\hat{Q}(\mathbf{s},\mathbf{a})-\hat{Q}(\mathbf{s},\mathbf{a})^{knn}||), (\mathbf{s},\mathbf{a})\sim\mathcal{D}_{\mathrm{online}}. \tag{20}$$

Given that the majority of our selected baselines are implemented using the double Q($\{Q_{\phi_1}, Q_{\phi_2}\}$), the offline pre-trained double Q can be readily utilized for the computation of intrinsic rewards, and we found that the performance of SERA is sutured when $\lambda$ is set to 1. We also provide a (Variance Auto Encoder) VAE implantation (Equation 21) of SERA, this realization is computing efficiency, but require extraly training a VAE model, due to Equation 2 won't require training thus we mainly test Equation 2.

$$r^{\mathrm{aug}}(\mathbf{s}, \mathbf{a}) = -\log p_{\hat{\phi}}(s|\hat{Q}(\mathbf{s},\mathbf{a})) = -\log \mathbb{E}_{z\sim q_\phi(z|\mathbf{s},\hat{Q}(\mathbf{s},\mathbf{a}))}\big[\frac{p_{\hat{\phi}}(\mathbf{s}|\hat{Q}(\mathbf{s},\mathbf{a}))}{q_\phi(z|\mathbf{s},\hat{Q}(\mathbf{s},\mathbf{a}))}\big], (\mathbf{s},\mathbf{a})\sim\mathcal{D}_{\mathrm{online}}. \tag{21}$$

We will test and compare the performance difference and computing efficiency between Equation 21 and Equation 20 in the future.

### D.4 CODEBASE

Our implementation is based on Cal-QL:`https://github.com/nakamotoo/Cal-QL`, VCSE:`https://sites.google.com/view/rl-vcse`. Additionally, we have included our source code in the supplementary material for reference. Readers can refer to our pseudocode (see Algorithm 1) for a comprehensive understanding of the implementation details. $\hat{Q}$ see [3].

---
**Algorithm 1** Training SERA

**Require:** Pre-collected data $\mathcal{D}_{\mathrm{offline}}$.
1: Initialize $\pi_\theta$, and $Q_{\phi_1}, Q_{\phi_2}$.
   // Offline Pre-training Stage.
2: **for** $k = 1, \cdots, K$ **do**
3:     Learn $Q_\phi$ on $\mathcal{D}_{\mathrm{offline}}$ by Equation 4 or 3 //We compute target Q value via $Q_{\mathrm{target}}$, learning $Q_{\mathrm{target}}$ by Empirical Momentum Average (EMA),*i.e.*, $Q_{\mathrm{target}} = (1-\alpha)Q_\phi + \alpha Q_{\mathrm{target}}$.
4:     Learn $\pi_\theta$ on $\mathcal{D}_{\mathrm{offline}}$ with Equation 5.
5: **end for**
   // Online Fine-tuning Stage.
6: **for** $k = 1, \cdots, K$ **do**
7:     Interacting $\pi_\theta$ to obtain $\mathcal{D}_{\mathrm{online}}$.
8:     Augmenting Reward in $\mathcal{D}_{\mathrm{online}}$ by Equation 1.
9:     Sample a batch offline data $\mathcal{D}_{\mathrm{offline}}$, and build training batch,*i.e.*, $\mathcal{D}_{\mathrm{mix}} = \mathcal{D}_{\mathrm{offline}} \cup \mathcal{D}_{\mathrm{online}}$ //mixture of offline and online is not necessary required, it depends on the quality of offline dataset.
10:     Learn $\pi_\theta, Q_{\phi_1}$, and $Q_{\phi_2}$ on $\mathcal{D}_{\mathrm{mix}}$ with the same objective in offline stage.
11: **end for**

---

### D.5 COMPUTING RESOURCES

Our experiments were run on a computer cluster with 4×32GB RAM, AMD EPYC 7742 64-Core CPU, and NVIDIA-A100 GPU, Linux. Most of our code base (The implantation of **Cal-QL**, **CQL**, **TD3+BC**, **SAC**) are based on JAX [4], part of our implantation (**IQL**, **AWAC**) are based on Pytorch[5] (We use different deep learning frameworks mainly to preliminary validate that our algorithm can work in various of deep learning frameworks).

### D.6 OUR HYPER-PARAMETER

**Hyper-parameter of SERA.** The K-nearest neighbors (knn) for SERA are configured as follows: [0, 10, 15, 25, 50, 85, 100, 110], and the parameter $\lambda$ in Equation 19 is set to 1.

---
[3] where $\phi_1$ and $\phi_2$ are the params of double Q Networks and $\hat{Q}(\mathbf{s},\mathbf{a}) = \min(Q_{\phi_1}(\mathbf{s},\mathbf{a}), Q_{\phi_2}(\mathbf{s},\mathbf{a}))$, and $x_i^{knn}$ is the $n_x(i)$-th nearest neighbor of $x_i$.
[4] `https://github.com/google/jax.git`
[5] `https://pytorch.org/`

**Hyper-parameter of Baselines** In the context of these algorithms, we conducted tests related to AWAC and IQL using the repository available at `https://github.com/tinkoff-ai/CORL`, while tests related to Cal-QL and CQL were performed using the repository accessible at `https://github.com/nakamotoo/Cal-QL`. The following five tables present fundamental but critical hyperparameter settings for five baseline algorithms.

Table 4: Hyper-parameters of AWAC.

| Hyperparameter | Value |
|---|---|
| 0ffline pre-train iterations | $1e^6$ |
| 0nline fine-tuning iterations | $1e^6$ |
| Buffer size | 20000000 |
| Batch size | 256 |
| learning rate | $3e^{-4}$ |
| $\gamma$ | 0.99 |
| awac $\tau$ | 5e-3 |
| awac $\lambda$ | 1.0 |
| Actor Architecture | 4× Layers MLP (hidden dim 256) |
| Critic Architecture | 4× Layers MLP (hidden dim 256) |

Table 5: Hyper-parameters of IQL.

| Hyperparameter | Value |
|---|---|
| 0ffline pre-train iterations | $1e^6$ |
| 0nline fine-tuning iterations | $1e^6$ |
| Batch size | 256 |
| learning rate of $\pi$ | $3e^{-4}$ |
| learning rate of V | $3e^{-4}$ |
| learning rate of Q | $3e^{-4}$ |
| $\gamma$ | 0.99 |
| IQL $\tau$ | 0.7 # *Coefficient for asymmetric loss* |
| $\beta$ (Inverse Temperature) | 3.0# *small beta → BC, big beta → maximizing Q* |
| Actor Architecture | 4× Layers MLP (hidden dim 256) |
| Critic Architecture | 4× Layers MLP (hidden dim 256) |

Table 6: Hyper-parameters of TD3+BC.

| Hyperparameter | Value |
|---|---|
| 0ffline pre-train iterations | $1e^6$ |
| 0nline fine-tuning iterations | $1e^6$ |
| learning rate of $\pi$ | $1e^{-4}$ |
| learning rate of Q | $3e^{-4}$ |
| $\gamma$ | 0.99 |
| Batch size | 256 |
| TD3 alpha | 2.5 |
| Actor Architecture | 4× Layers MLP (hidden dim 256) |
| Critic Architecture | 4× Layers MLP (hidden dim 256) |

Table 7: Hyper-parameters of Cal-QL. We only provide the basic setting, for more detail setting, please directly refer to `https://nakamotoo.github.io/projects/Cal-QL`

| Hyperparameter | Value |
| --- | --- |
| 0ffline pre-train iterations | $1e^6$ |
| 0nline fine-tuning iterations | $1e^6$ |
| learning rate of $\pi$ | $1e^{-4}$ |
| learning rate of Q | $3e^{-4}$ |
| $\gamma$ | 0.99 |
| Batch size | 256 |
| Actor Architecture | $4\times$ Layers MLP (hidden dim 256) |
| Critic Architecture | $4\times$ Layers MLP (hidden dim 256) |

Table 8: Hyper-parameters of CQL. CQL uses Cal-QL's code-base, and we only need to remove Cal-QL's calibration loss when deploying CQL.

| Hyperparameter | Value |
| --- | --- |
| 0ffline pre-train iterations | $1e^6$ |
| 0nline fine-tuning iterations | $1e^6$ |
| learning rate of $\pi$ | $1e^{-4}$ |
| learning rate of Q | $3e^{-4}$ |
| $\gamma$ | 0.99 |
| Batch size | 256 |
| Actor Architecture | $4\times$ Layers MLP (hidden dim 256) |
| Critic Architecture | $4\times$ Layers MLP (hidden dim 256) |

# E APPENDED EXPERIMENTAL RESULTS

In Table 9, we have provided completed offline-to-online results, including the ant-maze domain and the medium and medium-replay scenarios in the gym-mujoco environment, which is matched Figure 2. From Table 9, it can be observed that CQL paired with SERA exhibits the best average performance on the selected tasks. In Table 10, we compare a series of different efficient offline-to-

| Task | IQL | AWAC | TD3+BC | CQL | **CQL+SERA** | Cal-QL | **Cal-QL+SERA** |
| --- | --- | --- | --- | --- | --- | --- | --- |
| antmaze-large-diverse | 59 | 00 | 00 | 89.2 | 89.8 | 86.3 | **94.5** |
| antmaze-large-play | 51 | 00 | 00 | 91.7 | 92.6 | 83.3 | **95.0** |
| antmaze-medium-diverse | 92 | 00 | 00 | 89.6 | 98.9 | 96.8 | **99.6** |
| antmaze-medium-play | 94 | 00 | 00 | 97.7 | **99.4** | 95.8 | 98.9 |
| halfcheetah-medium | 57 | 67 | 49 | 69.9 | **87.9** | 45.6 | 46.9 |
| walker2d-meidum | 93 | 91 | 82 | 123.1 | **130.0** | 80.3 | 90.0 |
| hopper-medium | 67 | 101 | 55 | 56.4 | **62.4** | 55.8 | 61.7 |
| ant-medium | 113 | 121 | 43 | 123.8 | **136.9** | 96.4 | 104.2 |
| halfcheetah-medium-replay | 54 | 44 | 49 | 39.5 | **53.73** | 25.7 | 26.7 |
| walker2d-medium-replay | 90 | 73 | 90 | 87.6 | **107.65** | 7.4 | 29.7 |
| hopper-medium-replay | **91** | 56 | 88 | 4 | 15.0 | 3.5 | 1.8 |
| ant-medium-replay | 123 | **127** | 127 | 30 | 116.6 | 55.1 | 68.0 |
| **Average Fine-tuned** | 82.2 | 56.7 | 48.6 | 75.1 | 90.9 | 61.3 | 68.1 |

Table 9: Normalized score after online fine-tuning. We report the online fine-tuned normalized return. SERA obviously improves the performance of CQL and Cal-QL. In particular, CQL-SERA (mean score of **90.9**) is the best out of the 12 selected baselines. Notably, part of Antmaze's baseline results are *quoted* from existing studies. Among them, AWAC's results are *quoted* from Kostrikov et al. (2021) and CQL's results are *quoted* from Nakamoto et al. (2023).

online methods, including APL, PEX, and BR. Specifically, we tested these methods on the ant-maze domain and the `medium` and `medium-replay` tasks in the gym-mujoco environment. We found

that SERA shows the best overall performance, indicating that SERA, when paired with CQL, can achieve superior results.

| Task | CQL+APL | CQL+PEX | CQL+BR | CQL+SUNG | CQL+SERA |
|---|---|---|---|---|---|
| antmaze-large-diverse | 0 | 0 | 0.1 | 44.1 | 89.8 |
| antmaze-large-play | 0 | 0 | 0 | 52.7 | 92.6 |
| antmaze-medium-diverse | 36.8 | 0.3 | 13.6 | 85.6 | 98.9 |
| antmaze-medium-play | 22.8 | 0.3 | 22.2 | 86.3 | 99.4 |
| halfcheetah-medium | 44.7 | 43.5 | 56.7 | 79.7 | 87.9 |
| walker2d-meidum | 75.3 | 34.0 | 81.7 | 86.0 | 130.0 |
| hopper-medium | 102.7 | 46.3 | 97.7 | 104.1 | 62.4 |
| halfcheetah-medium-replay | 78.6 | 45.5 | 64.9 | 75.6 | 53.7 |
| walker2d-medium-replay | 103.2 | 40.1 | 88.5 | 108.2 | 107.7 |
| hopper-medium-replay | 97.4 | 66.5 | 78.8 | 101.9 | 15.2 |
| **Average Fine-tuned** | 56.2 | 27.6 | 50.4 | 82.4 | 83.8 |

Table 10: Comparison of various efficient offline-to-online methods.

## F    EXTENDED EXPERIMENTS

### F.1    TREND OF STATE ENTROPY CHANGING

**State Entropy as Intrinsic Reward.**    If the state density $\rho(\mathbf{s})$ is unknown, we can instead using non-parametric entropy estimator to approximate the state entropy (Seo et al., 2021). Specifically, given N i.i.d. samples $\{\mathbf{s}_i\}$, the k-nearest neighbors (knn) entropy estimator can be defined as[6]:

$$\hat{H}_N^k(S) = \frac{1}{N}\sum_{i=1}^{N}\log\frac{N\cdot||\mathbf{s}_i - s_i^{knn}||_2^{d_{\mathbf{s}}}\cdot\hat{n}_{\hat{\pi}}^{\frac{d_{\mathbf{s}}}{2}}}{k\cdot\Gamma(\frac{d_{\mathbf{s}}}{2}+1)} \propto \frac{1}{N}\sum_{i=1}^{N}\log||\mathbf{s}_i - \mathbf{s}_i^{knn}||. \tag{22}$$

**Visualization of State Entropy Changing.**    In this experiment, for each training step, we select the buffer and randomly sample 5000 instances to approximate the entropy using Equation 10. and then plot the trend of approximated state entropy. For the majority of the tasks, the state entropy of *AWAC-SERA* was either progressively greater than or consistently exceeded that of *AWAC-base*. This indicates that SERA effectively enhances the agent's exploratory tendencies, enabling them cover much more observation region.

Figure 9: The Changing of Approximated Entropy along with increasing training steps. We found that the approximated state entropy in the buffer collected by AWAC using SERA was greater in the later stages of online finetuning.

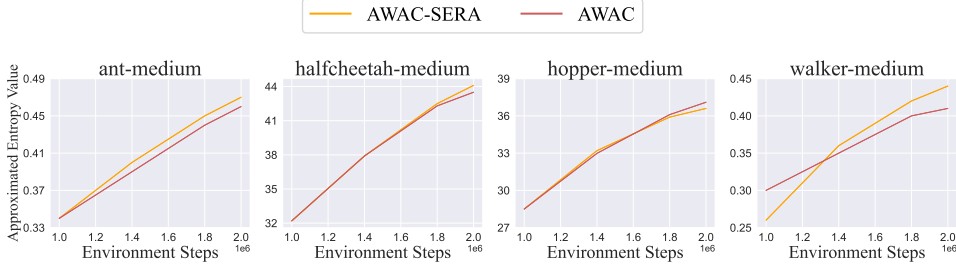

### F.2    PRETRAINED Q VS. RANDOM Q

**Pre-trained Q condition versus un-pre trained Q condition.**    To validate the statement in our main paper that intrinsic reward computation is influenced by the initialization of $Q$, we conducted experiments comparing the effects of pre-trained initialized $Q$ and from-scratch[7] trained $Q$ during

---

[6]$d_s$ is the dimension of state and $\Gamma$ is the gamma function, $\hat{n}_{\hat{\pi}} \propto 3.14$.

[7]We use from-scratch $Q$ to compute intrinsic reward, while continuing to utilize the offline-initialized $Q$ for conducting online fine-tuning.

intrinsic reward calculation. Our findings indicate that intrinsic rewards based on offline-initialized $Q$ generally outperform those derived from a from-scratch trained $Q$ across most tasks.

Figure 10: Offline Pre-trained Q condition vs. Randomly initialized Q condition. In the majority of our selected Gym-Mujoco tasks, the use of offline-initialized intrinsic reward conditions yielded better performance and higher sample efficiency. To provide clarity, *AWAC-base* means AWAC algorithm without any modification, *AWAC-SERA* signifies AWAC with SERA augmentation, and *AWAC-SERA (scratch)* denotes AWAC with SERA where the computation of reward conditions satisfying note 7

.

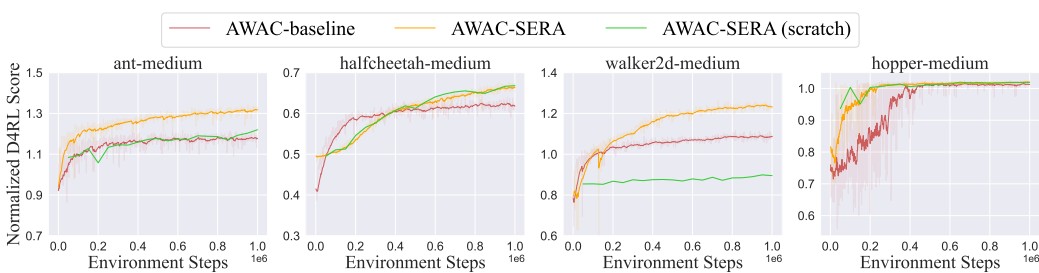

## F.3 Q CONDITION VS. V CONDITION

Differing from Kim et al. (2023), SERA conditions its intrinsic reward on $\min(Q_{\phi_1}(\mathbf{s}, \mathbf{a}), Q_{\phi_2}(\mathbf{s}, \mathbf{a}))$ rather than $V(\mathbf{s})$. In comparison to VCSE, SERA's advantage lies in its consideration of transitions. For example, assuming that there exist two transitions $T_1 = (\mathbf{s}, \mathbf{a}_1, \mathbf{s}_1)$ and $T_2 = (\mathbf{s}, \mathbf{a}_2, \mathbf{s}_2)$, since $T_1$ and $T_2$ have the same current observation, they will yield the same value conditioned intrinsic reward $-\log(\mathbf{s}|V(\mathbf{s}))$. This can introduce bias in the value learning process especially when current observation corresponds to a substantial number of valuable decisions and a limited number of low-value decisions. This is because low-value decisions can still receive relatively high intrinsic rewards based on the higher value expectations $V(\mathbf{s})$ for the current state, subsequently influencing the agent's decision-making. However, if we condition intrinsic reward on $Q(\mathbf{s}, \mathbf{a})$, it can take into account the decision-making simultaneously.

To further validate our claims, we chose AWAC as baseline and used both Q-network and V-network to compute the intrinsic reward's condition. We conducted tests on `halfcheetah-medium`, `walekr2d-medium`, `hopper-medium` and `ant-medium`. As shown in Figure 11, using the Q-network to compute condition has better performance compared to using V-network to compute condition.

Figure 11: Q condition vs. V condition. In this experiment, we selected AWAC as the base algorithm and compared using V network and Q network to calculate the intrinsic reward's condition. The experimental results indicate that using the Q-network to compute the condition leads to overall better performance for AWAC.

.

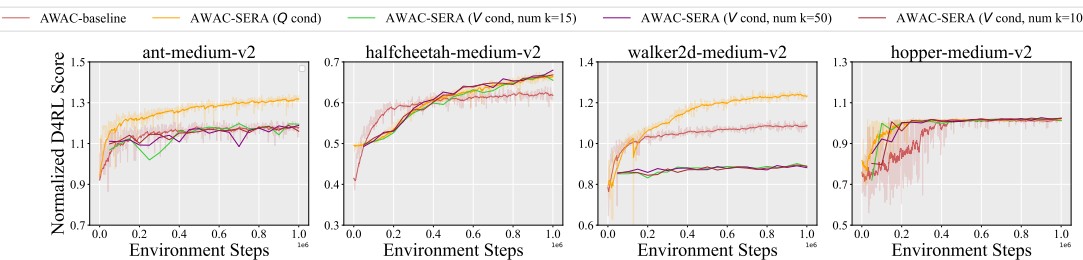

## G EXTENDED RELATED WORK

In this section, we systematically introduce recent developments in offline-to-online learning and summarize the corresponding methods,

The first perspective involves adopting a conservative policy optimization during online fine-tuning, typically achieved through the incorporation of policy constraints. Specifically, there are three main approaches within this category. The first approach constrains the predictions of the fine-tuning policy within the scope of offline support during online fine-tuning (Liu et al., 2023b). While this method contributes to achieving stable online fine-tuning performance, it tends to lead to overly conservative policy learning, and the accuracy of the estimation of offline support also influences the effectiveness of online fine-tuning. The second approach utilizes an offline dataset to constrain policy learning (Nair et al., 2021; Kostrikov et al., 2021; Xiao et al., 2023; Mark et al., 2023). However, the effectiveness of fine-tuning cannot be guaranteed if the dataset quality is poor. This method is sensitive to the quality of the dataset. The third approach employs pre-trained policies to constrain online fine-tuning, but this paradigm is influenced by the quality of the pre-trained policy (Zhang et al., 2023; Yu & Zhang, 2023).

The second perspective involves adopting a conservative approach during offline training, specifically using pessimistic constraints to learn Q to avoid OOD (Out-of-Distribution) issues. Research in this category primarily includes: Learning a conservative Q during offline pretraining and employing an appropriate experience replay method during online learning or using Q ensemble during offline pre-training to avoid OOD issues (Lee et al., 2021b; Lyu et al., 2022; Hong et al., 2023). However, as this approach introduces conservative constraints during critic updates, the value estimates between offline and online are not aligned, leading to a decrease in performance during early online fine-tuning. Therefore, Cal-QL introduces a calibrated conservative term to ensure standard online fine-tuning (Nakamoto et al., 2023).

Addtionally, there are also some other methods, such that ODT (Zheng et al., 2022) combined sequence modeling with Goal conditioned RL to conduct offline-to-online RL.

