\tag{5}
$$

In this equation, the term $V(\mathbf{s}_t) = \mathbb{E}_{\mathbf{a}_t \sim \pi}[Q(\mathbf{s}_t, \mathbf{a}_t) - \log \pi(\mathbf{a}_t \mid \mathbf{s}_t)]$ is defined.

**Lemma B.1 (Soft Policy Evaluation with SERA.)** *Given the modified soft bellman backup operator $\mathcal{T}_{sera}^\pi$ in Equation 5, along with a mapping $Q^0 : \mathcal{S} \times \mathcal{A} \to \mathbb{R}$ where $|\mathcal{A}| < \infty$. We define an iterative sequence as $Q^{k+1} = \mathcal{T}^\pi Q^k$. It can be shown that when index $k$ tends towards infinity, the sequence $Q^k$ converges to a soft Q-value of $\pi$.*

*proof.* Let us define the SERA reward as follows

$$
r_{sera}^\pi(\mathbf{s}_t, \mathbf{a}_t) \triangleq r(\mathbf{s}_t, \mathbf{a}_t) + \lambda \operatorname{Tanh}\left(\mathcal{H}\left(\mathbf{s}_t \mid \min\left(Q_{\phi_1}(\mathbf{s}_t, \mathbf{a}_t), Q_{\phi_2}(\mathbf{s}_t, \mathbf{a}_t)\right)\right)\right) + \mathbb{E}_{\mathbf{s}_{t+1} \sim p}\left[\mathcal{H}\left(\pi\left(\cdot \mid \mathbf{s}_{t+1}\right)\right)\right]
\tag{6}
$$

and rewrite the update rule as

$$
Q(\mathbf{s}_t, \mathbf{a}_t) \leftarrow r_{sera}^\pi(\mathbf{s}_t, \mathbf{a}_t) + \gamma \mathbb{E}_{\mathbf{s}_{t+1} \sim p, \mathbf{a}_{t+1} \sim \pi}[Q(\mathbf{s}_{t+1}, \mathbf{a}_{t+1})].
\tag{7}
$$

Then we can apply mathematical analysis of convergence for policy evaluation as outlined in Sutton & Barto (1998) to prove the result. It is essential to note that the assumption $|\mathcal{A}| < \infty$ is necessary to ensure the boundedness of the SERA reward."

**Lemma B.2 (Soft Policy Improvement with SERA)** *Let $\pi_{old} \in \Pi$, and let $\pi_{new}$ be the solution to the minimization problem defined as:*

$$
\pi_{\text{new}} = \arg \min_{\pi' \in \Pi} D_{\text{KL}}\left(\pi'(\cdot \mid \mathbf{s}_t) \left\| \frac{\exp(Q^{\pi_{\text{old}}}(\mathbf{s}_t, \cdot))}{Z^{\pi_{\text{old}}}(\mathbf{s}_t)}\right.\right).
\tag{8}
$$

*Then, it follows that $Q^{\pi_{new}}(\mathbf{s}_t, \mathbf{a}_t) \geq Q^{\pi_{old}}(\mathbf{s}_t, \mathbf{a}_t)$ for all $(\mathbf{s}_t, \mathbf{a}_t) \in \mathcal{S} \times \mathcal{A}$ provided that $|\mathcal{A}| < \infty$.*

*proof.* Starting from Equation 9, which has been established in the work by (Haarnoja et al., 2018), as:

$$
\mathbb{E}_{\mathbf{a}_t \sim \pi_{\text{new}}}[Q^{\pi_{\text{old}}}(\mathbf{s}_t, \mathbf{a}_t) - \log \pi_{\text{new}}(\mathbf{a}_t \mid \mathbf{s}_t)] \geq V^{\pi_{\text{old}}}(\mathbf{s}_t),