# OpenReview forum: "SERA: Sample Efficient Reward Augmentation in offline-to-online Reinforcement Learning"
_ICLR.cc/2024/Conference — Submitted to ICLR 2024_

### Official Review · Reviewer_NgYG · 2023-10-27

**Soundness:** 2 fair
**Presentation:** 2 fair
**Contribution:** 2 fair
**Rating:** 5
**Confidence:** 3

**Summary:**

This paper investigates sample-efficient offline-to-online reinforcement learning with reward augmentation technique. Specifically, this paper enhances VCSE with Q conditioned state entropy, deriving initially successful empirical findings on D4RL benchmark.

**Strengths:**

- The perspective of improving sample efficiency for offline-to-online RL seems interesting.

**Weaknesses:**

Overall, I think this paper does not meet the basic bar of ICLR, especially in terms of writing and experiments. I strongly suggest the authors proof-reading the paper thoroughly to make it a stronger submission. See detailed comments below.
- From Fig. 1, it seems that CQL-SERA > Cal-QL-SERA > Cal-QL baseline > CQL baseline, which contradicts to the empirical findings in Fig. 4.
- Is the unbiasedness of SERA theoretically guaranteed by replacing V function by Q function? If not, it is kind of over-claiming in Introduction.
- Please conduct sufficient research investigation on offline-to-online RL. A lot of related works are not appropriately referenced:

[1] Offline-to-online reinforcement learning via balanced replay and pessimistic q-ensemble, CoRL’22.

[2] Adaptive policy learning for offline-to-online reinforcement learning, AAAI’23.

[3] Policy Expansion for Bridging Offline-to-Online Reinforcement Learning, ICLR’23.

[4] Sample Efficient Offline-to-Online Reinforcement Learning, TKDE’23.

[5] Actor-Critic Alignment for Offline-to-Online Reinforcement Learning, ICML’23.

[6] Fine-tuning offline policies with optimistic action selection, NeurIPS workshop.

[7] A Simple Unified Uncertainty-Guided Framework for Offline-to-Online Reinforcement Learning, arXiv preprint.

[8] PROTO: Iterative Policy Regularized Offline-to-Online Reinforcement Learning, arXiv preprint.

[9] Ensemble-based Offline-to-Online Reinforcement Learning: From Pessimistic Learning to Optimistic Exploration, arXiv preprint.

[10] Towards Robust Offline-to-Online Reinforcement Learning via Uncertainty and Smoothness, arXiv preprint.

- Exploration has been discussed a lot by previous works on offline-to-online RL [3,4,6,7]. Please discuss advantages of SERA over them.
- In Section 3.1:

(1) $d_\mathcal{D}$ is not defined.

(2) Should not J(Q) be a MSE loss?

(3) In $\mathcal{B}_{\mathcal{M}}^{\pi}Q(s,a)$, the condition of the expectation is $s \sim \mathcal{D}$?

(4) Eq.(1) seems incorrect. Check Eq. (3.1) in Cal-QL paper.

(5) what is $s_i^{knn}$ in Eq.(2)?

- In Section 3.2:

(1) Eq.(4) seems incorrect. Please double-check.

(2) Overall, I cannot follow details in Section 3.2. Please provide step-by-step instructions in Appendix to make it more clear.

- In Section 4.1, Isn’t SERA a generic offline-to-online RL algorithm? Why the training objective is constrained to the framework of CQL and Cal-QL?
- Moreover, this paper claims to have an appendix pdf, but I cannot find the appendix in openreview.
- Why experiments are only conducted on 8 selected tasks. In general, MuJoCo has random/medium/medium-replay/medium-expert/etc. datasets. Consider these settings.
- It seems that there are only one random seed throughout the paper. Please repeat all the experiments with at least three different random seeds to control the randomness. Also, please report the mean and std value.
- Please consider more sufficient comparison in Fig. 5. Besides, in ant-medium, where is TD3+BC? In ant, halfcheetah, and walker2d, IQL seems performs better than IQL-SERA. Could you provide more explanations?
- Why only two tasks are selected in Fig. 6 (a)?
- Why only IQL is selected in Fig. 6 (b) on only two tasks?
- There are no sufficient ablation studies on each component of SERA. For example, you claim that condition on Q is better than V, thus, please derive some empirical findings to support this claim.
- Some typos:

(1) Reference format is not well-handled throughout the paper. In ICLR template: xxx (Author, et al., Year)

(2) In page 2: by maximizing no-conditioned -> non-conditioned; Anther reason -> Another.

(3) In page 3: some researches penalty the -> penalize; both offline and online RL., we -> delete the comma; improving Model-free offline-to-online RL -> model-free;

(4) In page 4: given N i.i.d samples -> $N$; consists of samples -> revise this sentence; Add , in Eq.(4); Equation. 4 -> Equation 4;

(5) In page 5: params -> parameters;

(6) In page 6: Differing -> Different;

**Questions:**

See weakness.

---

> ### Author Response · Authors · 2023-11-19
> **Response to Reviewer NgYG (Part 1, Overall responses)**
>
> $A1:$We thank the reviewers for the analysis of the initial version of the manuscript. There are significant differences between the current version and the initial version. We have addressed most of the issues based on the comparison between the two versions of the manuscript. Additionally, in Part 1, we will provide overall responses to select questions. In the subsequent parts, we will reorganize the reviewer's questions and provide detailed responses.
>
> $Q2:$ Should not J(Q) be a MSE loss?
>
> $A2:$ Many thanks to the reviewers for pointing out the shortcomings. It was a writing error.
>
> $Q3:$In $\mathcal{B}_{M}^{\pi}Q(s,a)$, the condition of the expectation is $s\sim \mathcal{D}$?
>
> $A3:$$\mathcal{B}$ is bellman operator, and $\mathcal{B}_{M}^{\pi}Q(s,a)=r(s,a)+\gamma\times Q(s',\pi(s'))$ where $(s,a,s')\sim \mathcal{D}$
>
> $Q4:$ Overall, I cannot follow details in Section 3.2. Please provide step-by-step instructions in Appendix to make it more clear.
>
> $A4:$ The latest version has reorganized Section 3.2 in the main text and provided relevant proofs
>
> $Q5:$In Section 4.1, Isn’t SERA a generic offline-to-online RL algorithm? Why the training objective is constrained to the framework of CQL and Cal-QL?
> Moreover, this paper claims to have an appendix pdf, but I cannot find the appendix in openreview.
> Why experiments are only conducted on 8 selected tasks. In general, MuJoCo has random/medium/medium-replay/medium-expert/etc. datasets. Consider these settings.
> It seems that there are only one random seed throughout the paper. Please repeat all the experiments with at least three different random seeds to control the randomness. Also, please report the mean and std value.
>
> $A5:$ Firstly, our theorem 4.1 proves that SERA can ensure the policy optimization of soft Q, so we tend to test it on algorithms based on soft Q, thus we choose Cal-QL and CQL to conduct the test. Secondly, in Figure 4, we tested the performance of SERA combined with various model-free algorithms (TD3+BC, AWAC, IQL), and the improvement brought by SEAR is particularly evident for AWAC. Additionally, we used AWAC as the base algorithm, and in Figure 5, we compared the performance differences between various exploration algorithms (RND, SE) and SEAR. SERA performs the best overall. Additionally, the latest version we submitted has added an appendix and expanded the D4RL experiments to $\texttt{medium-replay}$.
>
> $Q6:$ Please consider more sufficient comparison in Fig. 5. Besides, in ant-medium, where is TD3+BC? In ant, halfcheetah, and walker2d, IQL seems performs better than IQL-SERA. Could you provide more explanations?
>
> $A6:$ Firstly, IQL combined with SERA generally achieves slightly better performance in most cases. We provide the reason for the poor performance of IQL on $\texttt{walker2d-medium}$. Beta in IQL is a crucial hyperparameter, where a small beta makes the training target close to behavioral cloning, and a large beta makes the training target close to Q learning. However, when testing SERA, we did not intentionally adjust the hyperparameters of various algorithms. Therefore, a too-small beta may limit the exploratory nature of SERA, thereby restricting the potential improvement in sample efficiency brought by SERA.
>
> $Q7:$ Why only IQL is selected in Fig. 6 (b) on only two tasks?
>
> $A7:$ In the new version, we have added AWAC and compared the performance of IQL and AWAC when combined with RND, SE, and SERA on these two tasks. SERA performs the best.
>
> $Q8:$ There are no sufficient ablation studies on each component of SERA. For example, you claim that condition on Q is better than V, thus, please derive some empirical findings to support this claim.
>
> $A8:$ The current version has expanded with more ablation experiments compared to the initial version (section.5.2). In the next few days, we will further expand necessary ablation experiments based on the experimental progress, such as comparing Q condition with V condition.
>
> $Q9:$ Some typos $\cdots$
>
> $A9:$ Thanks for the reviewer's suggestions. We have submitted a new version of the manuscript and will systematically address the details based on the writing advice from the reviewers.

---

> > ### Comment · Reviewer_NgYG · 2023-11-21
> >
> > Thank you for your feedback. While I acknowledge considerable enhancements in the writing of this paper, it's important to note that my concerns have not been addressed in a point-by-point manner. Therefore, I have opted to maintain my original score.

---

> ### Author Response · Authors · 2023-11-21
> **Response to Reviewer NgYG (Part 2: New extended and ablation studies )**
>
> We thank the reviewer's question of the deficiencies in our initial version of the ablation experiments. In the response of Part 2, we will carefully organize and answer the ablation study-related question and provide more detailed descriptions of the various ablation and extended experimental results we have added in the new version manuscript.
>
> ---
> ## Extended experiments
>
> $Q1:$ It seems that there are only one random seed throughout the paper. Please repeat all the experiments with at least three different random seeds to control the randomness. Also, please report the mean and std value.
>
> $A1:$ In Table 1, we report results from at least three repeated experiments, with multiple runs tested for each repetition. The new experimental results include both the mean and variance. To facilitate simultaneous presentation, we have included the results below.
>
> |Task Name| CQL | CQL+SERA | CalQL | CalQL+SERA
> |:---------:|:---------:| :---------:|:---------:|:---------:|
> | antmaze-large-diverse |89.2  |89.8$\pm$3.2|86.3$\pm$0.2|94.5$\pm$1.7|
> | antmaze-large-play    |91.7  |&92.6$\pm$ 1.3|83.3$\pm$9.0|95.0$\pm$1.1|
> | antmaze-medium-diverse|89.6 |98.9$\pm$0.2|96.8$\pm$1.0 |99.6$\pm$0.1|
> | antmaze-medium-play   |97.7 |99.4$\pm$0.4|95.8$\pm$0.9|	98.9$\pm$0.6|
> | halfcheetah-meidum    | 69.9  |87.9$\pm$2.3|45.6$\pm$0.0|46.9$\pm$0.0|
> | walker2d-medium       |123.1 |130.0$\pm$0.0 |80.3$\pm$0.4 |90.0$\pm$3.6|
> | hopper-medium         |56.4|62.4$\pm$ 1.3|55.8$\pm$0.7|61.7$\pm$2.6|
> | ant-medium            |123.8|136.9$\pm$1.6|96.4$\pm$0.3|104.2$\pm$3.0|
>
> Additionally, in Figure 3, we refer to [1] to conduct a statistical analysis of Table 1, and the improvement brought by SERA is statistically significant.
>
> $Q2:$ Why only two tasks are selected in Fig. 6 (a)?
>
> $A2:$ We thank the reviewer's questions. Figure 6(a) serves as further verification of whether SERA can be applied to soft-Q-based algorithms. We have conducted extensive experiments based on CQL-SAC and Cal-QL, which also demonstrate that SERA can be used in conjunction with Soft-Q.
>
> $Q3:$ Why only IQL is selected in Fig. 6 (b) on only two tasks?
>
> $A3:$ In the new version, we have added AWAC and compared the performance of IQL and AWAC when combined with RND, SE, and SERA on these two tasks in Figure 5 (b). SERA performs the best.
>
> $Q4:$ Why experiments are only conducted on 8 selected tasks.
>
> $A4:$ We have supplemented the experimental results for medium-replay in Figure 1 (training curve) and Table 9 (fine-tuned results).
>
> $\text{Comparisons 1}:$ We choose CQL as base algorithm, and compare APL, PEX, BR, SERA in Figure 7 and Table 10, we also provide average performance below:
>
> | CQL+APL | CQL+PEX | CQL+BR | CQL+SERA
> |:---------:| :---------:|:---------:|:---------:|
> |56.2 |27.6  |50.4|83.8 |
>
> ---
>
> ## New Ablation
>
> $Q1:$ Is the unbiasedness of SERA theoretically guaranteed by replacing V function by Q function? If not, it is kind of over-claiming in Introduction. $Q2:$For example, you claim that condition on Q is better than V, thus, please derive some empirical findings to support this claim.
>
> $A1,A2:$ Thank you for pointing out the shortcomings. Our previous manuscript indeed provided overly affirmative descriptions on this matter (Q condition is better than V condition). Therefore, in the supplementary Appendix (F.3), we have added additional experiments to compare the performance differences between Q condition and V condition. Specifically, In Appendix F.3, we chose AWAC as the base algorithm and added a comparison of SERA based on Q condition and V condition to calculate intrinsic rewards. We found that, overall, Q condition performs better than V condition.
>
> $\text{Pre-trained Q vs. Random Initialized Q when conduct SERA: }$ In the Figure 10 (Appendix F2), we added new ablation experiments using AWAC as the base algorithm on $\texttt{hopper-medium}$, $\texttt{walker-medium}$, $\texttt{halfcheetah-medium}$, and $\texttt{ant-medium}$ tasks. Specifically, we compared the performance of SERA when using pretrain-Q networks and randomly initialized Q networks to calculate intrinsic rewards, and found that using pre-trained Q networks for intrinsic reward computation yields better performance than using un-pretrained Q networks, thus Q condition is effective, and more carefully analysis, please see the response to Reviewer zoZX (Question.1)
>
> [1] Rishabh Agarwal, et.al. Deep reinforcement learning at the edge of the statistical precipice, 2022

---

> ### Author Response · Authors · 2023-11-21
> **Response to Reviewer NgYG (Point by point manner, Part 1/3)**
>
> Thanks for actively participating in the rebuttal process and providing numerous suggestions, and we believe the current version has addressed most of the reviewers' concerns. To facilitate communication and streamline responses, we will reply point-by-point under this section (total three part). Additionally, we have added a new section to categorize and organize responses to different kinds of issues. Thanks a lot.
>
> ---
>
> $Q1:$ From Fig. 1, it seems that CQL-SERA > Cal-QL-SERA > Cal-QL baseline > CQL baseline, which contradicts to the empirical findings in Fig. 4.
>
> $A1:$ We thank the reviewer's question. Firstly, this is a schematic diagram, and its primary purpose is to illustrate the training phase of SERA. Additionally, we will adjust the positions of the curves based on the final average experimental results.
>
> $Q2:$ Is the unbiasedness of SERA theoretically guaranteed by replacing V function by Q function? If not, it is kind of over-claiming in Introduction.
>
> $A2:$ Thank you for pointing out the shortcomings. Our previous manuscript indeed provided overly affirmative descriptions on this matter (Q condition is better than V condition). Therefore, in the supplementary Appendix (F.3), we have added additional experiments to compare the performance differences between Q condition and V condition. Specifically, In Appendix F.3, we chose AWAC as the base algorithm and added a comparison of SERA based on Q condition and V condition to calculate intrinsic rewards. We found that, overall, Q condition performs better than V condition.
>
> $Q3:$ Please conduct sufficient research investigation on offline-to-online RL. A lot of related works are not appropriately referenced:
>
> $A3:$ We thank reviewer for pointing out the shortcomings. The referenced papers are indeed crucial recent studies. However, due to limited space in the main text, we prefer to introduce studies most directly related to our research.  If necessary, we may choose to add a section on advances in offline-to-online learning in the supplementary material.
>
> $Q4:$ Exploration has been discussed a lot by previous works on offline-to-online RL [3,4,6,7]. Please discuss advantages of SERA over them.
>
> $A4:$ From the experimental results (average results on gym-mujoco (medium, medium-replay) and antmaze), our method outperforms PEX[3], BR[.1], APL[.2], SUNG[7]
>
> | CQL+APL | CQL+PEX | CQL+BR |  CQL+SUNG|CQL+SERA
> |:---------:| :---------:|:---------:|:---------:|:---------:|
> |56.2 |27.6  |50.4|82.4 |83.8|
>
> Next, we will compare SERA with [3, 4, 6, 7] in terms of methods and theories:
>
> - Compared to PEX[3], SERA does not require training an additional policy, and experimental results show that SERA outperforms PEX.
>
> - Compared to [4], SERA does not incorporate meta-adaptation into the framework. Instead, it focuses solely on enhancing offline-to-online performance from the perspective of exploration. Therefore, in theoretical terms, this represents an essential and unique approach.
>
> - SERA and [6] represent two completely opposite approaches. SERA adopts an aggressive exploration strategy during online training, while [6] aims to explore only slightly beyond the replay buffer during online fine-tuning. Therefore, intuitively, SERA can collect a more diverse dataset more quickly.
>
> - SERA and SUNG[7] share a commonality in that SUNG tends to favor optimistic exploration strategies, while SERA also encourages exploration by the agent. The difference lies in the fact that SERA does not require additional training, and SUNG's VAE may be influenced by data quality during pretraining. Additionally, experimental results show that SERA slightly outperforms SUNG
>
> $Q5:$ typos(1) to (5); Q6: Errors of Equation:
>
> We thank the reviewers for pointing out the shortcomings in our writing. The latest version of our manuscript has shown significant improvement compared to the initial version, and we will further enhance our expression.

---

> ### Author Response · Authors · 2023-11-21
> **Response to Reviewer NgYG (Point by point manner, Part 2/3)**
>
> $Q6:$ In Section 4.1, isn't SERA a generic offline-to-online RL algorithm? Why the training objective is constrained to the framework of CQL and Cal-QL?
>
> $A6:$ We thank the reviewer's suggestions. In the main text, we have expanded the experiments to popular algorithms such as TD3+BC, SAC, and AWAC. SERA shows overall good performance across these algorithms.
>
> $Q9:$ Moreover, this paper claims to have an appendix pdf, but I cannot find the appendix in openreview.
>
> The supplementary material has been added after the main text. Specifically,
>
> - In Appendix B, we provide some theoretical proofs relevant to this paper. Specifically,  Appendix B.1 explains the connection between State entropy maximization and exploration. Appendix B.2 contains Theorems 4.1 and 4.2. Theorem 4.1 ensures policy improvement in Soft-Q algorithms when combined with SERA.
>
> - In Appendix C$\sim$D, we have systematically organized the implementation details of SERA, including implementation, computing resources and hyperparameters.
>
> - In Appendix E, We have added the performance of the algorithm on medium-replay, and further comparisons were made between SERA and various efficient offline-to-online algorithms.
>
> $Q10:$ Why experiments are only conducted on 8 selected tasks. In general, MuJoCo has random/medium/medium-replay/medium-expert/etc. datasets. Consider these settings.
>
> $A10:$ We have supplemented the experimental results for medium-replay in Figure 1 (training curve) and Table 9 (fine-tuned results).
>
> $Q11:$ It seems that there are only one random seed throughout the paper. Please repeat all the experiments with at least three different random seeds to control the randomness. Also, please report the mean and std value.
>
> $A11:$ In Table 1, we report results from at least three repeated experiments, with multiple runs tested for each repetition. The new experimental results include both the mean and variance. To facilitate simultaneous presentation, we have included the results below.
>
> |Task Name| CQL | CQL+SERA | CalQL | CalQL+SERA
> |:---------:|:---------:| :---------:|:---------:|:---------:|
> | antmaze-large-diverse |89.2  |89.8$\pm$3.2|86.3$\pm$0.2|94.5$\pm$1.7|
> | antmaze-large-play    |91.7  |&92.6$\pm$ 1.3|83.3$\pm$9.0|95.0$\pm$1.1|
> | antmaze-medium-diverse|89.6 |98.9$\pm$0.2|96.8$\pm$1.0 |99.6$\pm$0.1|
> | antmaze-medium-play   |97.7 |99.4$\pm$0.4|95.8$\pm$0.9|	98.9$\pm$0.6|
> | halfcheetah-meidum    | 69.9  |87.9$\pm$2.3|45.6$\pm$0.0|46.9$\pm$0.0|
> | walker2d-medium       |123.1 |130.0$\pm$0.0 |80.3$\pm$0.4 |90.0$\pm$3.6|
> | hopper-medium         |56.4|62.4$\pm$ 1.3|55.8$\pm$0.7|61.7$\pm$2.6|
> | ant-medium            |123.8|136.9$\pm$1.6|96.4$\pm$0.3|104.2$\pm$3.0|
> |average                |92.7 | 94.7        | 78.8       | 86.4        |
>
> $Q12:$ Please consider more sufficient comparison in Fig. 5. Besides, in ant-medium, where is TD3+BC? In ant, halfcheetah, and walker2d, IQL seems performs better than IQL-SERA. Could you provide more explanations?
>
> $A12:$ We thank the reviewer's suggestions. Firstly, IQL combined with SERA generally achieves slightly better or the same performance in most cases. In particular, we provide the reason for the poor performance of IQL on $\texttt{walker2d-medium}$. $\beta$ in IQL is a crucial hyperparameter, where a small $\beta$ makes the training target close to behavioral cloning, and a large $\beta$ makes the training target close to Q learning. However, when testing SERA, we did not intentionally adjust the hyperparameters of various algorithms. Therefore, a too-small $\beta$ may limit the exploratory nature of SERA, thereby restricting the potential improvement in sample efficiency brought by SERA.

---

> ### Author Response · Authors · 2023-11-21
> **Response to Reviewer NgYG (Point by point manner, Part 3/3)**
>
> $Q13:$ Why only two tasks are selected in Fig. 6 (a)?
>
> $A13:$ We thank the reviewer's questions. Figure 6(a) serves as further verification of whether SERA can be applied to soft-Q-based algorithms (theorem 4.1 has the guarantee of soft-Q optimization). We have conducted extensive experiments based on CQL-SAC and Cal-QL, which demonstrate that SERA can be used in conjunction with Soft-Q.
>
> $Q14:$ Why only IQL is selected in Fig. 6 (b) on only two tasks?
>
> $A14:$ In the new version, we have added AWAC and compared the performance of IQL and AWAC when combined with RND, SE, and SERA on these two tasks. SERA performs the best.
>
> $Q15:$ There are no sufficient ablation studies on each component of SERA. For example, you claim that condition on Q is better than V, thus, please derive some empirical findings to support this claim.
>
> $A15:$We thank the reviewer's suggestions, and we have incorporated a substantial number of ablation experiments in the latest version of the manuscript. We have also systematically organized the previous experimental results in the $\textbf{Response to Reviewer NgYG (Part 2: New extended and ablation studies )}$ section, facilitating the reviewers in their examination and comparative analysis.
>
> ---
>
> $Q16:$ Some typos: $\cdots$
>
> $A16:$ We thank reviewers for pointing out these shortcomings. Your feedback is very helpful in improving our writing.
>
> [.1] Lee S, Seo Y, Lee K, et al. Offline-to-online reinforcement learning via balanced replay and pessimistic q-ensemble[C]//Conference on Robot Learning. PMLR, 2022: 1702-1712.
>
> [.2] Adaptive policy learning for offline-to-online reinforcement learning, AAAI’23.

---

> ### Author Response · Authors · 2023-11-23
> **Dear Reviewer NgYG, we kindly and sincerely invite you to provide further feedback.**
>
> Dear Reviewer  NgYG,
>
> As the rebuttal stage is nearing its conclusion, we kindly invite you to provide feedback on our latest manuscript and the current responses. Thank you.
>
> Best regards.

---

> > ### Comment · Reviewer_NgYG · 2023-11-23
> >
> > Thanks for your comprehensive response addressing each point individually. However, I still harbor reservations about certain concerns, particularly Q1, Q3, Q6, and Q11. Additionally, I'd like to highlight that the substantial revision and response, especially near the author-reviewer discussion deadline, indeed place a significant burden on the reviewers. Despite these considerations, I am raising my score to 5.

---

> > > ### Author Response · Authors · 2023-11-23
> > >
> > > We appreciate the reviewer's continued active participation in our discussions and responses to our questions. Thank you very much. We will do our best to address Q1, Q3, Q6, and Q11 before the rebuttal deadline.

---

> > > > ### Author Response · Authors · 2023-11-23
> > > >
> > > > Once again, thanks for your suggestions and active participation in our discussions. These suggestions are very helpful in improving the quality of our manuscript. Thank you!

---

> ### Author Response · Authors · 2023-11-23
> **Further discussion of Q1 and Q3**
>
> Further discussion of Q1
>
> In our new manuscript, we have revised Figure 1, which now serves two main purposes: 1) Figure 1 succinctly illustrates the process of SERA. 2) Figure 1 demonstrates that SERA can benefit certain algorithms.
>
> ---
>
> Further discussion of Q3: We supplement related work in this section, and we have added such content in the Appendix of new manuscript.
>
> In the past, efforts have primarily focused on enhancing offline-to-online performance from two perspectives.
>
> The first perspective involves adopting a conservative policy optimization during online fine-tuning, typically achieved through the incorporation of policy constraints. Specifically, there are three main approaches within this category. The first approach constrains the predictions of the fine-tuning policy within the scope of offline support during online fine-tuning [1]. While this method contributes to achieving stable online fine-tuning performance, it tends to lead to overly conservative policy learning, and the accuracy of the estimation of offline support also influences the effectiveness of online fine-tuning. The second approach utilizes an offline dataset to constrain policy learning [2,3,4,5]. However, the effectiveness of fine-tuning cannot be guaranteed if the dataset quality is poor. This method is sensitive to the quality of the dataset. The third approach employs pre-trained policies to constrain online fine-tuning, but this paradigm is influenced by the quality of the pre-trained policy [6,7].
>
> The second perspective involves adopting a conservative approach during offline training, specifically using pessimistic constraints to learn Q to avoid OOD (Out-of-Distribution) issues. Research in this category primarily includes: Learning a conservative Q during offline pretraining and employing an appropriate experience replay method during online learning or using Q ensemble during offline pretraining to avoid OOD problems [8,10,11]. However, as this approach introduces conservative constraints during critic updates, the value estimates between offline and online are not aligned, leading to a decrease in performance during early online fine-tuning. Therefore, Cal-QL introduces a calibrated conservative term to ensure standard online fine-tuning [9].
>
> Addtionally, there are also some other methods, such that ODT[12] combined sequence modeling with Goal conditioned RL to conduct offline-to-online RL.
>
> reference:
>
> (Policy constrain)
>
> [1] Supported Policy Optimization for Offline Reinforcement Learning. (SPOT)
>
> [2] AWAC: Accelerating Online Reinforcement Learning with Offline Datasets. (AWAC)
>
> [3] Offline Reinforcement Learning with Implicit Q-Learning. (IQL)
>
> [4] The In-Sample Softmax for Offline Reinforcement Learning. (InAC)
>
> [5] ine-Tuning Offline Policies With Optimistic Action Selection. (O3F)
>
> [6] Policy Expansion for Bridging Offline-to-Online Reinforcement Learning. (PEX)
>
> [7] Actor-Critic Alignment for Offline-to-Online Reinforcement Learning. (ACA)
>
> (Pessimitic critic)
>
> [8] Offline-to-Online Reinforcement Learning via Balanced Replay and Pessimistic Q-Ensemble .
>
> [9] Cal-QL: Calibrated Offline RL Pre-Training for Efficient Online Fine-Tuning.
>
> [10] Mildly conservative q-learning for offline reinforcement learning.
>
> [11] Confidence-conditioned value functions for offline reinforcement learning.
>
> (Additional method)
>
> [12] Online Decision Transformer.

---

### Official Review · Reviewer_zoZX · 2023-10-28

**Soundness:** 2 fair
**Presentation:** 1 poor
**Contribution:** 2 fair
**Rating:** 5
**Confidence:** 4

**Summary:**

This paper studies the problem of fine-tuning pre-trained offline RL agents. Specifically, it proposed a reward augmentation framework, named Sample Efficient Reward Augmentation (SERA), to encourage exploration in the fine-tuning stage with Q conditional state entropy. SERA further uses state marginal matching (SMM) and penalizes OOD state actions. Experiments on the D4RL benchmark tasks showed the proposed SERA outperformed other baselines.

**Strengths:**

- This paper investigates an important question in offline RL.
- The proposed method outperformed other baseline in the D4RL benchmark task.

**Weaknesses:**

- Firstly, the writing is not good enough. Many sentences are not rigorous or confusing. For example:
    - In the first paragraph, "such paradigm can only learn similar or slightly better performance than behavioural policy" is not true. Because model-based offline RL methods can sometimes significantly improve the performance w.r.t. the behavioural policy.
    - In the third paragraph, "The second approach employs offline RL with policy regression". What does the "policy regression" mean? Or it's a typo of "policy regularization".
    - "underestimate the value of the offline buffer in comparison to the ground truth returns" => should be "underestimate the value of OOD samples in the offline buffer"

- There are too many typos and grammar errors:
    1. "some researches penalty the Q values" ==> penalize
    2. Missing period after "or implicitly regularize the bellman equation"
    3. "It similarly train agent" ==> trains
    4. "high sampling efficiency" ==> sample
    5. extra period "on both offline and online RL., we "
    6. "as a Markov decision Process" ==> Decision
    7. "A denotes the actions space" ==> action
    8. missing comma in "tau = {s0, a0, r0, ..., st, at rt}"
    9. missing "the" in "in offline-to-online RL problem setting"
    10. "Bellman equation iteration"  ==> "Bellman iteratio equation"
    11. "it always suffer from" ==> suffers
    12. missing norm notation in the one step Bellman equation
    13. missing right bracket in "if (s', pi(\cdot | s') \notin D"
    14. "to penalty the OOD state actions" ==> penalize
    15. "expected regression" ==> expectile regression
    16. "by rollout behavioural policy" ==> unrolling
    17. "thus has the" ==> having
    18. "only maximize" ==> maximizing
    19. "rather E[H[s]]" ==> rather than
    20. "where Tanhs see" ==> sees

- There are some missing SOTA baselines for offline-to-online fine-tuning in the experiments: Reincarnating RL [1], PEX [2], InAC [3]

[1] (Agarwal et al., NeurIPS' 22) Reincarnating reinforcement learning: Reusing prior computation to accelerate progress

[2] (Zhang et al., ICLR' 23) Policy Expansion for Bridging Offline-to-Online Reinforcement Learning

[3] (Xiao et al., ICLR' 23) The In-Sample Softmax for Offline Reinforcement Learning

**Questions:**

- "which is unbiased in the early online process" => why it's unbiased?

- Since the main argument of this work is a new exploration method for fine-tuning offline RL agents. I think it should compare to other  intrinsic reward baselines, i.e, state entropy, RND, ICM.

---

> ### Author Response · Authors · 2023-11-19
> **Response to Reviewer zoZX (Part of Questions)**
>
> $Q1:$ "which is unbiased in the early online process" => why it's unbiased?
>
> $A1:$  We thank the essiential question of Reviewer, and we would like to further clarify our use of 'un-biased' in reviwer's comments.  Such 'un-biased' are not related to the concept of distribution shift between offline and online. Instead, such 'un-based' are specifically referring to whether the Q used in calculating the state entropy of the Q condition is pre-trained. In VCSE [.1], the authors considered a purely online setting and used a value network with random initialization to compute the condition of value-conditioned entropy. However, a value network with random initialization may not accurately estimate the values of all states in the early stages of online training. Consequently, when calculating the value-conditioned state entropy, there may be situations where states with different true values correspond to the same output from the value network.
>
> Therefore, we use a value network with random initialization to calculate the value-conditioned entropy as a 'biased' prediction in the early stages of online training. Such biased prediction doesn't align with the original intention of VCSE because low-value states and high-value states may be assigned the same value condition in the early stages of online training, it could result in states from low-value regions being included in high-value regions, or states from high-value regions being included in low-value regions.
>
> However, we can't immediately conclude that an offline pre-trained Q network is necessarily better than a randomly initialized Q network. There is a distribution shift issue between offline and online. Therefore, we chose AWAC as the base algorithm and combined with SERA (pretrained Q condition), SERA (from-scratch Q condition) to test on various offline-to-online tasks such as $\texttt{ant-medium}$, $\texttt{hopper-medium}$, $\texttt{walker-medium}$, and $\texttt{halfcheetah-medium}$. As shown in Figure 10 (Appendix F.2), using an offline-pretrained Q as a condition performs better compared to using a randomly initialized Q as a condition and consistently outperforms the baseline. In particular, using a randomly initialized Q as a condition even achieves lower performance than the baseline in $\texttt{walker-medium}$. These experimental results all demonstrate the advantage of using a pre-trained Q network when computing conditions
>
> $Q2:$ Since the main argument of this work is a new exploration method for fine-tuning offline RL agents. I think it should compare to other intrinsic reward baselines, i.e, state entropy, RND, ICM.
>
> $A2:$ Thanks for the valuable suggestions from the reviewers. We chose AWAC and IQL as base algorithms, combined with SERA, RND, and state entropy (SE), and conducted offline-to-online testing on $\texttt{walker-medium}$ and $\texttt{hopper-medium}$. As shown in Figure 5 (b), we found that the algorithm combined with SERA performs the best.
>
> [.1] Dongyoung Kim, et al. Accelerating reinforcement learning with value-conditional state entropy exploration, 2023

---

> > ### Author Response · Authors · 2023-11-21
> > **Response to Reviewer zoZX (Part of Weakness)**
> >
> > $Q1:$ Firstly, the writing is not good enough. Many sentences are not rigorous or confusing. ;$Q2:$ There are too many typos and grammar errors
> >
> > $Q1,Q2:$ We thank Reviewer for the series of writing suggestions. In the latest version we submitted, we have systematically organized the entire writing and we will further improve the expression of this paper.
> >
> > $Q3:$ There are some missing SOTA baselines for offline-to-online fine-tuning in the experiments.
> >
> > $A3:$ We thank Reviewer's suggestion to add new comparison experiments (Compared with previous efficient algorithms). As shown in Figure 7 (new manuscript) and Table 10 (new manuscript) in the Appendix, CQL-SERA outperforms CQL-APL, CQL-PEX, and CQL-BR on (antmaze, and gym-mujoco (medium, medium-replay) domains). At the same time, for convenience, we have provided the average performance of the compared algorithms on all tasks below:
> >
> > | CQL+APL | CQL+PEX | CQL+BR | CQL+SERA
> > |:---------:| :---------:|:---------:|:---------:|
> > |56.2 |27.6  |50.4|83.8 |

---

> > ### Author Response · Authors · 2023-11-23
> > **Dear Reviewer zoZX, we kindly and sincerely invite you to provide further feedback, thank you**
> >
> > Dear Reviewer zoZX,
> >
> > As the rebuttal stage is nearing its conclusion, we kindly invite you to provide feedback on our latest manuscript and the current responses. Thank you.
> >
> > Best regards.

---

### Official Review · Reviewer_sGsS · 2023-10-30

**Soundness:** 2 fair
**Presentation:** 1 poor
**Contribution:** 2 fair
**Rating:** 3
**Confidence:** 5

**Summary:**

This paper focuses on offline-to-online RL and proposes improving the performance by enhancing exploration during online fine-tuning with a reward augmentation framework, SERA. The intrinsic rewards are calculated by implementing State Marginal Matching (SMM) and penalizing out-of-distribution (OOD) state actions.

**Strengths:**

- The proposed method is easy to understand.
- The technique seems sound.

**Weaknesses:**

- See the questions.

**Questions:**

==Major concerns==
- The authors are strongly advised to revise this paper carefully. There are so many typos in this paper, which affects the normal comprehension of this paper.
- I do not understand how to calculate Equation (3) in practice when the state is high dimensional continuous variables. Can the authors provide the analysis?
- What is the relation between Equation (2) and Equation (4) when calculating the critic-conditioned intrinsic reward?
- Every intrinsic reward calculation must be calculated by KNN, so the efficiency of physic time consumption may be a little poor.
- I can not find the appendix mentioned in this paper.
- Can the author provide the whole comparisons about D4RL datasets?
- The format of some citations is wrong.
- What about the random seed in the experiments?

==Minor concerns==
- The authors should explain all symbols that appear in this paper, e.g., in Section 3.1, the authors do not introduce $d_D(.|s)$ and $\mathcal{G}_{\mathcal{M}}$.
- In Definition 1, why the critic-conditioned entropy does not contain “-”. Besides, if there are N states that are used for calculating the critic conditioned entropy,
- In Equation (3), the initial state distribution is $\rho_0(S)$, but in Section 3.1, the initial state distribution is defined as $p(s_0)$. Besides,
- In Equation (4), the symbols of the left side and the right side are very different. Can the authors provide a detailed derivation?
- The authors should provide the derivation about “Another reason is that maximizing Es∼ρ(s)[Hπ[s]] is equivalent to minimize DKL(ρπ(s)||p∗(s)) thus has the mathematical guarantee.”
- Different reference expressions about figures.



==Typos==
- Section 3.1 “Model-free Offline RL”: “In particular, Model-free”-> “In particular, model-free”
- Section 3.1 “Model-free Offline RL”: “Specifically, Model-free” -> “Specifically, model-free”
- Section 3.1 “Model-free Offline RL”: “one step bellman equation i.e. …. which” -> “one step bellman equation, i.e. xxxx, which”
- Section 3.1 “Model-free Offline RL”: “Previous studies have extensively studied such a problem, such that CQL was proposed to penalty the OOD state actions by conservative term (Equation 1), and IQL implicitly learns Q function with expected regression without explicit access to the value estimation of OOD state-actions.”
- Section 3.1: “state entropy(Seo et al., 2021)” -> “state entropy (Seo et al., 2021)”
- Section 3.1: “i.i.d”-> ““i.i.d.””
- Section 3.2: grammatical mistake: “Specifically, we first use the offline methods to …..”
- Section 3.2: “\pi_{beta}” -> “\pi_{\beta}”
- Section 3.2: “Equation. 4” -> “Equation (4)”
- Section 3.2: “SMM,i.e.” -> “SMM, i.e.”
- Section 3.2: “Only maximize” -> “Only maximizing”
- Section 4.1: “… are the params of double Q Networks” -> “… are the parameters of double Q Networks”
- Section 4.1: “in addition to testing SERA” -> “in addition to test SERA”
There are so many typos, so I suggest the authors check this paper carefully.

---

> ### Author Response · Authors · 2023-11-19
> **Response to reviewer sGsS (Part of Major Concerns)**
>
> $Q1:$ The authors are strongly advised to revise this paper carefully. There are so many typos in this paper, which affects the normal comprehension of this paper.
>
> $A1:$ We appreciate the valuable suggestions provided by the reviewers. The current version differs significantly from the initial version. Specifically,
>
> - (1) in Section 3 and 4, we have redefined the fundamental concepts required for this paper, and we have provided a mathematical proof (Appendix B.1 and B.2) demonstrating that SERA can be used in conjunction with Soft-Q. Thus, the experimental performance of SERA can be interpreted.
>
> - (2) In Section 5, we extended experiments related to d4rl from medium to medium-replay (Figure 1, Table 9). Additionally, we reported the averaged results of experiments conducted multiple times in Table 1. Furthermore, in Figure 3, we conducted a statistical analysis of Table 1, and the statistical results indicate that the improvement brought by SERA to the algorithm is significant.
>
> - (3) In Section 6, we have included a substantial number of ablation experiments to comprehensively validate the effectiveness of SERA from multiple perspectives. These ablation experiments include: in Appendix F.3, the performance of SERA under V condition is better than under Q condition, which confirms the point made in the last paragraph of Section 3. Additionally, the comparison between pre-train Q and un-pretrain Q, as added in the supplementary pages, attests that SERA can effectively utilize the Q network pre-trained offline to calculate intrinsic rewards.
>
> $Q2:$ I do not understand how to calculate Equation (3) in practice when the state is high dimensional continuous variables. Can the authors provide the analysis?
>
> $A2:$ Thanks for reviewer's question. Firstly, although this formula is in discrete form, it is independent of whether the variables are continuous or not. Additionally, this formula calculates entropy for a batch and does not cover all samples. However, with a sufficient number of training iterations, it is equivalent to maximizing the state entropy over the entire space.
>
> $Q3:$ What is the relation between Equation (2) and Equation (4) when calculating the critic-conditioned intrinsic reward?
>
> $A3:$ Thanks for the reviewer's suggestions. Firstly, we have removed Equation 2 in the main text and systematically defined the concept and mathematical form of state entropy under the joint probability distribution (new version manuscript, Definition 2). We have also provided the implementation form of Q-conditioned state entropy (new version manuscript, Equation 2). In addition, based on the initial version, Formula 2 in the initial version cannot directly calculate Q conditioned state entropy. The initial version manuscript adopts the same method as the current version to calculate Q-conditioned state entropy, which is same as Equation 2 in new version manuscript.
>
> $Q4:$ Every intrinsic reward calculation must be calculated by KNN, so the efficiency of physic time consumption may be a little poor.
>
> $A4:$ We thank the reviewer's question. Firstly, the main advantage of mathematically approximating entropy is that it does not require training. Secondly, in the new version, we have supplemented the calculation method based on VAE (Equation 21). We plan to compare the speed and cost between the VAE-based and mathematical methods in the future.
>
> $Q5:$ I can not find the appendix mentioned in this paper.
>
> $A5:$ The current version includes the supplementary material directly appended to the end of the main text. (introduction see Response to Reviewer NgYG (Point by point manner, Part 2/3), Q9)
>
>
> $Q6:$ Can the author provide the whole comparisons about D4RL datasets?
>
> $A6:$  We have added experimental results for "medium-replay" in Figure 2 and Table 9. Additionally, we refer [1.] to perform statistical analysis on the results from Table 1 in Figure 3. SERA brings significant improvements to Cal-QL and CQL.
>
> $Q7:$ The format of some citations is wrong.
>
> $A7:$ We thank the reviewer for pointing out the shortcomings of writing. The current version of the writing has seen significant improvement compared to the initial version. The suggestions from the reviewers have been instrumental in enhancing the quality of the manuscript.
>
> $Q8:$ What about the random seed in the experiments?
>
> $A8:$ Thanks, we have presented the mean and variance of the final performance for each task in Table 1 (new version), we have included the results below (see: Response to Reviewer NgYG (Part 2: New extended and ablation studies )).

---

> ### Author Response · Authors · 2023-11-21
> **Response to reviewer sGsS (Part of Minor Concerns)**
>
> $Q1:$ in Section 3.1, the authors do not introduce and In Definition 1, $Q2:$why the critic-conditioned entropy does not contain “-” $\cdots$.
>
> $A1,A2:$ We thank suggestions provided by the reviewer, and we believe these questions and recommendations will further refine our definitions and writing. In the new manuscript, we have organized and redefined several concepts essential for understanding this paper. Some symbols and previously defined content have been removed, but we still provide explanations for certain symbols used previously. $\mathcal{G}$ means the greedy operator,ie, $\pi \leftarrow argmax_{\pi} E_{s\sim D}[Q(\pi(s),s)]$, and $d(\cdot|s)$ denotes state-marginal distribution in dataset $D$.
>
> $Q3:$ The authors should provide the derivation about “Another reason is that maximizing $E_{s\sim\rho(s)}[H_{\pi}[s]]$ is equivalent to minimize $D_{KL}(\rho_{\pi}(s)||p^*(s))$.”
>
> $A3:$ Good questions! In the new version paper, we have systematically defined and demonstrated that maximizing entropy serves two main purposes (Appendix.B.1): (1) Maximizing state entropy encourages the agent to explore the entire state space, enabling the acquisition of states not initially included in the dataset through increased exploration. (2) Maximizing state entropy establishes a trade-off, where one aspect aids in approximating any target density. Furthermore, our Theorem 4.1 ensures that, when combined with SERA in soft Q algorithms, policy improvement is guaranteed. Certainly, we acknowledge that in the past, our expression regarding this matter may have been too affirmative. Merely maximizing entropy cannot guarantee the absolute achievement of SMM (State Marginal Matching). Achieving SMM in an absolute sense requires combining term 3, ie, $\max E_{s\sim \rho_{\pi}}[\log p^*(s)]$. This term can offset term1 in the Appendix.B.1 (page 16, Equation.7), thereby achieving SMM.
>
> [1] Rishabh Agarwal, et.al. Deep reinforcement learning at the edge of the statistical precipice, 2022

---

> ### Author Response · Authors · 2023-11-21
> **Response to Reviewer sGsS (Part of Typos)**
>
> We thank the reviewers for pointing out the shortcomings in our initial manuscript. The suggestions from the reviewers are very helpful in improving the quality of our manuscript. The current version has shown significant improvement in writing compared to the initial version.

---

> ### Author Response · Authors · 2023-11-23
> **Dear Reviewer sGsS, we kindly and sincerely invite you to provide further feedback, thank you**
>
> Dear Reviewer sGsS,
>
> As the rebuttal stage is nearing its conclusion, we kindly invite you to provide feedback on our latest manuscript and the current responses. Thank you.
>
> Best regards.

---

### Official Review · Reviewer_pDE4 · 2023-10-31

**Soundness:** 3 good
**Presentation:** 3 good
**Contribution:** 3 good
**Rating:** 6
**Confidence:** 4

**Summary:**

This paper introduces a generalized reward enhancement framework known as SERA, which aims to boost online fine-tuning performance by designing intrinsic rewards, thereby improving the online performance of offline pre-trained policies. SERA achieves this by implicitly enforcing state marginal matching and penalizing out-of-distribution state behaviors, encouraging the agent to cover the target state density, resulting in superior online fine-tuning outcomes. Experimental results consistently demonstrate the effectiveness of SERA in enhancing the performance of various algorithms in offline-to-online settings.

**Strengths:**

1. The exploration of the offline-to-online problem in this study holds great relevance and is imperative for practical implementations, aligning seamlessly with the demands of real-world situations.
2. The fundamental idea at the core of this study is firmly grounded. While the concept presented in this paper is rather straightforward, involving the introduction of an exploration strategy during the online phase to enhance performance, the specific exploration technique employed is quite novel and has demonstrated favorable results in the experiments.

**Weaknesses:**

1. In the experimental section, the author conducted experiments solely on the medium dataset in MuJoCo. However, according to the consensus in the field of offline-to-online research, it is generally recommended to perform experiments on at least three types of datasets: medium, medium-replay, and medium-expert, in order to validate the effectiveness of the method.
2. The method proposed in this paper is primarily an extension of CQL and Cal-QL. However, in the context of the offline-to-online field, the actual compared baselines are limited to AWAC and Cal-QL. It is advisable for the authors to consider comparing their method with other more efficient algorithms such as Balanced Replay[1], PEX[2], and ENOTO[3].
3. The SERA algorithm, proposed in this paper, primarily enhances online performance by designing intrinsic rewards to encourage exploration. This concept has been mentioned in previous works such as O3F[4] and ENOTO, although SERA employs different exploration methods. While introducing exploration during the online phase can enhance performance, it may introduce another challenge: instability due to distribution shift, which can lead to performance degradation in the early stages of online learning. This issue has been discussed in many offline-to-online works and is a critical metric in this field. However, it might not be very evident on the medium dataset. Therefore, the authors should consider conducting additional experiments on the medium-replay and medium-expert datasets to verify whether performance degradation occurs.
4. In Figure 4, the experimental results for the Antmaze environment are challenging to discern, as the curves for various algorithms are intertwined and unclear. The author should consider optimizing the representation of these experimental results for better clarity.
5. In Table 1, only the mean values of the algorithm results are presented, with a lack of information regarding the errors or variances associated with these results.

[1] Lee S, Seo Y, Lee K, et al. Offline-to-online reinforcement learning via balanced replay and pessimistic q-ensemble[C]//Conference on Robot Learning. PMLR, 2022: 1702-1712.

[2] Zhang H, Xu W, Yu H. Policy Expansion for Bridging Offline-to-Online Reinforcement Learning[J]. arXiv preprint arXiv:2302.00935, 2023.

[3] Zhao K, Ma Y, Liu J, et al. Ensemble-based Offline-to-Online Reinforcement Learning: From Pessimistic Learning to Optimistic Exploration[J]. arXiv preprint arXiv:2306.06871, 2023.

[4] Mark M S, Ghadirzadeh A, Chen X, et al. Fine-tuning offline policies with optimistic action selection[C]//Deep Reinforcement Learning Workshop NeurIPS 2022. 2022.

**Questions:**

See weakness part.

---

> ### Author Response · Authors · 2023-11-19
> **Response to Reviewer pDE4**
>
> $Q1:$ In the experimental section, the author conducted experiments solely on the medium dataset in MuJoCo. However, according to the consensus in the field of offline-to-online research, it is generally recommended to perform experiments on at least three types of datasets: "medium", "medium-replay", and "medium-expert", in order to validate the effectiveness of the method.
>
> $A1:$ We thank the reviewer for the suggestion to supplement the experiments (D4RL). In the new version of the manuscript, we have added tests on "medium-replay" in D4RL. As shown in Table.9 (SERA improve CQL 21%, Cal-QL 12.2%), and CQL-SERA has the best overall performance (Compared with various baselines IQL, AWAC, TD3+BC, CQL, Cal-QL).
>
> $Q2:$ The method proposed in this paper is primarily an extension of CQL and Cal-QL. However, in the context of the offline-to-online field, the actual compared baselines are limited to AWAC and Cal-QL. It is advisable for the authors to consider comparing their method with other more efficient algorithms such as Balanced Replay, PEX, and ENOTO.
>
> $A2:$ Good suggestions! We have supplemented relevant tests. As shown in Figure 7 (new manuscript) and Table 10 in the appendix (new manuscript), CQL-SERA outperforms CQL-APL, CQL-PEX, and CQL-BR on (antmaze, and gym-mujoco {medium, medium-replay} domains). At the same time, for convenience, we have provided the average performance of the compared algorithms on all tasks below:
>
> | CQL+APL | CQL+PEX | CQL+BR | CQL+SERA
> |:---------:| :---------:|:---------:|:---------:|
> |   56.2   |    27.6   |   50.4  |  83.8     |
>
> $Q3:$ The SERA algorithm, proposed in this paper, primarily enhances online performance by designing intrinsic rewards to encourage exploration. This concept has been mentioned in previous works such as O3F[4] and ENOTO, although SERA employs different exploration methods. While introducing exploration during the online phase can enhance performance, it may introduce another challenge: instability due to distribution shift, which can lead to performance degradation in the early stages of online learning. This issue has been discussed in many offline-to-online works and is a critical metric in this field. However, it might not be very evident on the medium dataset. Therefore, the authors should consider conducting additional experiments on the medium-replay and medium-expert datasets to verify whether performance degradation occurs
>
> $A3:$ Nice suggestions! In question 1, we supplemented experiments related to D4RL. Additionally, in the supplementary pages, we provided mathematical proofs demonstrating that SERA ensures policy improvement during soft Q optimization (Theorem B.3 in Appendix.B.2) and encourages the agent to uniformly explore the entire observation space (Appendix.B.1 "Why does ASMM encourage covering the target density?"). Additionally, [.2] explains that diverse data can address the shortcomings of conservative policies, and SERA can encourage the agent to explore the environment as much as possible, facilitating the collection of more diverse dataset, therefore, from a theoretical perspective, SERA also helps improve various soft Q-based algorithms.
>
> $Q4:$ In Figure 4 (Figure 2 in new manuscript), the experimental results for the Antmaze environment are challenging to discern, as the curves for various algorithms are intertwined and unclear. The author should consider optimizing the representation of these experimental results for better clarity.
>
> $A4:$ Good suggestions! According to the reviewer's suggestion, we further optimized this figure using uniform sampling. The distinguishability between curves has significantly improved compared to the previous version. The trends of the curves remain consistent with the statements in the paper, showing that CQL-SERA and Cal-QL-SERA converge faster and achieve better fine-tuned results.
>
> $Q5:$ In Table 1, only the mean values of the algorithm results are presented, with a lack of information regarding the errors or variances associated with these results.
>
> $A5:$ Thanks to the reviewer's suggestion, we have reported the results of experiments conducted in at least three repetitions, with multiple runs for each repetition. We have presented the mean and variance of the final performance for each task in Table 1. Additionally, we employed the statistical methods described in [.1] and further analyzed the experimental results from Table 1 in Figure 3. The results demonstrate that SERA significantly enhances the performance of CQL and Cal-QL (higher median, IQM, and mean scores are better).
>
> [.1] Rishabh Agarwal, et.al. Deep reinforcement learning at the edge of the statistical precipice, 2022
>
> [.2] Yicheng Luo, et.al. Finetuning from offline reinforcement learning: Challenges, trade-offs and practical solutions, 2023.

---

> > ### Comment · Reviewer_pDE4 · 2023-11-20
> >
> > Thanks for your detailed feedback and improvement to the paper. I believe most of my concerns are addressed and I have raised my score to 6.

---

> > > ### Author Response · Authors · 2023-11-20
> > >
> > > Thank you, and we appreciate your valuable suggestions for this study.

---

### Author Response · Authors · 2023-11-11
**Modification about the new paper version**

We would like to express our gratitude to all the reviewers for their valuable suggestions on our initial manuscript. We strongly believe that these suggestions will help us to refine our algorithm and significantly improve the overall quality of this manuscript. Based on the reviewers' feedback, we have identified two main shortcomings in the first version of the PDF: first, the writing needs to be improved, and second, there is a lack of experiments demonstrating the effectiveness of SERA.

Fortunately, we have been actively improving our research since submitting it to ICLR24 and have already incorporated most of the points raised by the reviewers into our new version. Due to a large number of changes, we are not highlighting the changes in colour.

+ We have taken the reviewers' feedback seriously in order to improve the quality and clarity of our work. As a result, we have essentially rewritten the entire manuscript, including the Introduction, Methods, Experiments, and Appendix sections:
- Experimental aspect：
    + We extended the tests from gym-mujoco to medium-replay (Figure 1, Table 9). Meanwhile, we have supplemented the statistical analysis in Fig. 3, which provides evidence of the significant improvement brought by our method.
    + We added a comparison between SERA and various past efficient algorithms (PEX, APL, etc.)
    + We added a comparison between SERA and different exploration methods (RND, SE)
    + We added a comparison between Q condition and V condition, where Q condition outperformed V condition.
    + We have expanded our ablation experiments by comparing pre-trained and non-pre-trained Q as conditions. We also quantified the state entropy. These experimental results are consistent with some of our theoretical claims (Section 5.2 and Appendix E.1 and E.2).
- Theoretical aspect：
    + We have redefined all fundamental concepts and reorganized the writing of the paper.
    + In the supplementary material, we provide a mathematical analysis (Appendix B.1) that demonstrates the effectiveness of encouraging exploration through the maximum entropy approximation. In addition, we prove that SERA can ensure the improvement of the policy optimization for soft Q-based methods(Appendix B.2, theorm 4.1). Furthermore, it can guarantee conservative policy improvements when using double Q (Appendix B.2, theorm 4.2).
The revised manuscript now provides a more comprehensive and coherent account of our research, ensuring that the paper delivers a stronger and more impactful message.

+ To further strengthen the empirical foundation of our study, we have conducted additional experiments based on your valuable suggestions. These new experiments address certain aspects that were not adequately covered in the original submission. We believe that these additions contribute significantly to the overall robustness and reliability of our findings.

---
Because of our huge revision, we strongly suggest that the reviewers revisit our new submission. Once again, we express our gratitude for the valuable suggestions provided by the reviewers. Thank you!

---

### Meta-Review · Area_Chair_qpeH · 2023-12-12

**Metareview:**

### Summary
The paper presents a novel approach to offline-to-online RL called SERA, aimed at enhancing the performance of offline-to-online reinforcement learning. By introducing intrinsic rewards to promote exploration during online fine-tuning, SERA addresses the limitations of direct fine-tuning of pre-trained policies. It achieves this by implicitly enforcing state marginal matching and penalizing out-of-distribution state actions, leading to improved coverage of target state density and superior online fine-tuning results. The paper demonstrates SERA's consistent effectiveness across various offline algorithms in offline-to-online scenarios, showcasing its versatility and potential for significant performance enhancement.

### Decision
Overall, The authors have done a tremendous job addressing the issues the reviewers raised. The initial reviewers were quite negative about this paper, but during the rebuttal period, some reviewers increased their scores after the authors' rebuttal. One of the major issues with this paper is the clarity and writing, which confused some of the reviewers. The initial version of the paper seems rushed; the authors tried to address some of these issues. As it stands right now, in terms of writing, I do believe this paper still doesn't meet the ICLR's bar. I would recommend the authors take their time to revise their paper to make it easier to read and resubmit to a different venue.

**Justification For Why Not Higher Score:**

The initial version of the paper was very rushed and poorly written. The authors improved but still doesn't meet ICLR's bar.

**Justification For Why Not Lower Score:**

N/A

---

### Decision · Program_Chairs · 2024-01-16

Reject